# LCA: LOCAL CLASSIFIER ALIGNMENT FOR CONTINUAL LEARNING

**Tung Tran**[1]**, Danilo Vasconcellos Vargas**[1]**, Khoat Than**[2]*

[1]Kyushu University, Fukuoka, Japan    [2]Hanoi University of Science and Technology, Hanoi, Vietnam
`tran.tung.son.949@s.kyushu-u.ac.jp`
`vargas@inf.kyushu-u.ac.jp`
`khoattq@soict.hust.edu.vn`[†]

## ABSTRACT

A fundamental requirement for intelligent systems is the ability to learn continuously under changing environments. However, models trained in this regime often suffer from catastrophic forgetting. Leveraging pre-trained models has recently emerged as a promising solution, since their generalized feature extractors enable faster and more robust adaptation. While some earlier works mitigate forgetting by fine-tuning only on the first task, this approach quickly deteriorates as the number of tasks grows and the data distributions diverge. More recent research instead seeks to consolidate task knowledge into a unified backbone, or adapting the backbone as new tasks arrive. However, such approaches may create a (potential) *mismatch* between task-specific classifiers and the adapted backbone. To address this issue, we propose a novel *Local Classifier Alignment* (LCA) loss to better align the classifier with backbone. Theoretically, we show that this LCA loss can enable the classifier to not only generalize well for all observed tasks, but also improve robustness. Furthermore, we develop a complete solution for continual learning, following the model merging approach and using LCA. Extensive experiments on several standard benchmarks demonstrate that our method often achieves leading performance, sometimes surpasses the state-of-the-art methods with a large margin.

## 1 INTRODUCTION

In real-world scenarios, data arrives continuously, requiring deployed models to adapt to evolving distributions while preserving previously learned knowledge. This challenge, known as the stability–plasticity dilemma in continual learning, captures the difficulty of balancing adaptation with retention. A widely adopted benchmark approximates this setting by partitioning a dataset into disjoint tasks and training a model sequentially without access to earlier data (Wang et al., 2024). At test time, the model must handle all tasks without being given the task identity. This setup is referred to as *Class-Incremental Learning* (CIL) (Van de Ven et al., 2022).

Pre-trained models (PTMs) have recently emerged as a strong foundation for this setting, in contrast to earlier methods that trained networks from scratch (Li & Hoiem, 2017; Kirkpatrick et al., 2017). PTMs can be obtained through supervised or self-supervised training, and include large multimodal models such as CLIP (Radford et al., 2021) as well as vision backbones like the Vision Transformer (Dosovitskiy et al., 2020). Their broad generalization ability makes them effective feature extractors, requiring only lightweight adaptation and thereby reducing forgetting.

Nevertheless, naive sequential adaptation still leads to degradation on past tasks. Restricting updates to the first task avoids forgetting but blocks useful transfer to later ones. Since each task introduces unique information, the final model must capture both shared and task-specific components. A natural approach is to incrementally adapt on each task and then merge the resulting models into a unified backbone. Insights from Linear Mode Connectivity and the Lottery Ticket Hypothesis

---

*Corresponding author.
†Code is available at: this https URL

(Frankle et al., 2020) suggest that task-specific solutions can be connected through low-loss paths and rely on sparse, critical parameters, making them amenable to combination. Advances in model merging (Ilharco et al., 2022; Yadav et al., 2023) further support this strategy, with both theoretical guarantees (Li et al., 2025) and empirical evidence on adaptation tasks (Akiba et al., 2025) showing that merged models can preserve and even enhance knowledge. Thus, it is reasonable to treat each task-specific model as a standalone expert containing complementary knowledge, whose consolidation yields a stronger overall backbone. However, merging backbones across tasks introduces a new challenge: classifiers trained independently may no longer align with the integrated backbone. Because these classifiers cannot be retrained without access to past data, even small parameter shifts can lead to severe drops in performance on earlier tasks. Addressing this misalignment is the central focus of our work.

Our contributions in this work are as follows:

- We introduce a novel loss, *Local Classifier Alignment* (LCA), for aligning CIL classifiers. This loss not only well aligns the backbone with the classifier, but also ensures robustness of the classifier. It also can contribute to reducing overlap between classes and hence improving classifier's performance.
- We provide theoretical analysis that decomposes the test error of a CIL model into three fundamental parts, including feature distribution shift, class-wise loss, and robustness. Those parts must be well controlled to assure a high performance for all observed tasks. Such a theory is crucial to support reliability of LCA and trustworthy CIL.
- We propose a complete CIL solution in which model merging (for PEFT parameters only) is used to slightly adapt the backbone to new tasks and LCA serves as the main alignment loss to reduce mismatches between classifiers and the backbone. In our method, each class is represented as a Gaussian in the feature space, and LCA jointly optimizes all classifiers.
- We conduct extensive experiments on seven benchmark datasets. Our results show that LCA consistently improves performance over baselines, enhances robustness under diverse scenarios, and can be integrated with other CIL methods to further boost their effectiveness. Figure 1 shows superiority of LCA on seven benchmark datasets.

## 2 RELATED WORKS

**Representation-Based Methods.** This line of research leverages the representational power of large pre-trained models for continual learning (Wang et al., 2024). Trained on massive and diverse datasets, these models provide strong transferability and inherent robustness against forgetting. One prominent direction adapts prompting techniques from natural language processing, where prompts are modeled as learnable parameters attached to the inputs (Wang et al., 2022c;b; Tran et al., 2025). These prompts act as additional instructions that guide model predictions during continual learning. Another direction adapts the pre-trained backbone only once during the first task by using parameter-efficient fine-tuning (PEFT) modules and then relies on class prototypes for inference (Panos et al., 2023; Zhou

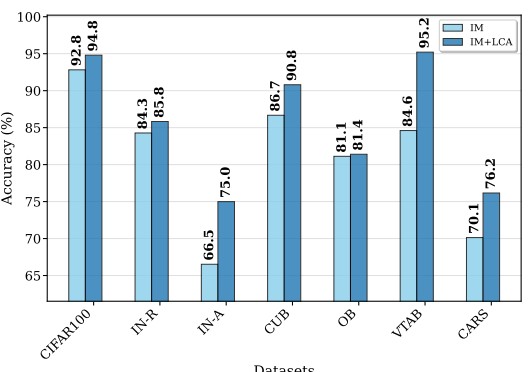

Figure 1: A comparison between IM and IM+LCA. IM is the result after only done the Incremental Merging step, while IM+LCA has Local Classifier Alignment as the last step.

et al., 2025; McDonnell et al., 2023; Perez et al., 2018). Such prototype-based approaches classify via cosine similarity can avoid forgetting because accumulated prototypes across tasks are tasks order-invariant. However, empirical evidence shows that adaptation only in the first task is insufficient, since data distributions in real-world scenarios vary substantially across tasks, which limits this method's applicability in long training horizons.

**Incremental Backbone Evolution.** A complementary line of research updates the backbone throughout the task sequence. For example, SLCA (Zhang et al., 2023) reduces forgetting by apply-

ing a smaller learning rate to the backbone than to the classifier, while methods such as MagMax (Marczak et al., 2024), EASE (Zhou et al., 2024), and MOS (Sun et al., 2025b) focus on integrating task-specific components into a unified backbone. In all cases, past classifiers are typically frozen to avoid bias toward the current task's data, which inevitably creates a mismatch between the evolving backbone and the fixed classifiers. To mitigate this issue, EASE reweights old classifiers using semantic similarity between new and old prototypes, while MOS dynamically selects suitable backbone adapters at inference time. Another line of research seeks to enrich past prototypes through augmentation, as in FeTrIL (Petit et al., 2023), PASS (Zhu et al., 2021b), IL2A (Zhu et al., 2021a), and CCFA (Kim et al., 2024), or to enhance the capacity of current classifiers through boosting, as in FOSTER (Wang et al., 2022a). Our approach retrains all classifiers after the unification step using samples drawn from simple Gaussian models, together with a novel loss that emphasizes local robustness. This procedure reduces the mismatch between the backbone and the classifiers and improves the stability of the overall model.

**Model Merging.** Another related direction is model merging, which has recently gained attention for constructing a unified model from independently trained task-specific ones. Early methods such as FisherMerging (Matena & Raffel, 2022) use the Fisher Information Matrix to weight parameter importance, while Task Arithmetic (Ilharco et al., 2022) represents task updates as vectors, enabling explicit addition or subtraction of knowledge. More advanced approaches like TIES-Merge (Yadav et al., 2023) address task interference by resolving sign conflicts across task vectors. Although these methods show strong performance in out-of-domain generalization (Rame et al., 2022), applying them directly to continual learning often introduces a mismatch between the merged backbone and fixed classifiers, and incurs storage overhead since parameters from all past tasks must be retained. In contrast, our approach builds on the spirit of model merging by incrementally consolidating PEFT modules, which reduces memory requirements, and coupling this with a continuous training scheme that solidifies accumulated knowledge across tasks.

# 3 METHODOLOGY

This section presents our methodology for continual learning, which consists of two complementary components. The first focuses on incrementally merging task-specific backbones into a unified model, while maintaining proximity across tasks by initializing new training from the latest merged backbone. The second addresses the mismatch that arises when frozen task-specific classifiers interact with the consolidated backbone, introducing an alignment mechanism based on class-wise local regions. Together, these components enable the model to effectively retain past knowledge while adapting to new tasks.

## 3.1 PROBLEM FORMULATION

In class-incremental learning (CIL), the objective is to train a single model sequentially on a series of (potentially infinite number of) tasks. Each task $i$ is associated with a dataset

$$\mathcal{D}_i = \{(x, y) \mid x \in \mathcal{X}_i, \ y \in \mathcal{Y}_i\},$$

where $x$ denotes an input sample and $y$ its corresponding label. Here, $\mathcal{X}_i$ represents the input space and $\mathcal{Y}_i$ the label space for task $i$. A defining characteristic of the CIL setting is that the label spaces of different tasks are strictly disjoint:

$$\forall i \neq j, \quad \mathcal{Y}_i \cap \mathcal{Y}_j = \emptyset.$$

This assumption implies that each new task introduces a set of novel classes that the model has not encountered before. Consequently, the model must continuously expand its knowledge while preserving performance on previously learned classes, making the prevention of catastrophic forgetting a central challenge in CIL.

There are different strategies for constructing classifiers in continual learning, such as Nearest Class Mean (NCM) classifiers, which assign labels by comparing test features to stored class prototypes. In this work, however, we focus on the more general and widely used setup of progressively expanding Multi-Layer Perceptrons (MLPs) on top of the feature extractor. Instead of training a single unified classifier over all classes, a new MLP head is added for each task, with its output dimension matching the number of classes introduced by that task. This approach not only reduces storage but

---

**Algorithm 1** Incremental Merging (IM)

---

**Input:** Datasets $\{\mathcal{D}_1, \ldots, \mathcal{D}_T\}$, pretrained model $\theta_{\text{pretrained}}$, base PEFT $\theta_{\text{peft}_0}$
**Output:** Merged PEFT module $\theta_{\text{merged}}$

  1:   $\tau \leftarrow \mathbf{0}$                                                            ▷ Task vector accumulator
  2: **for** $i = 1$ to $T$ **do**                                     ▷ Sequentially process tasks
  3:      $\theta_{\text{peft}_i} \leftarrow \texttt{finetune}(\theta_{\text{peft}_{i-1}}, \mathcal{D}_i)$                     ▷ Task-$i$ adaptation
  4:      $\tau_{\text{curr}} \leftarrow \theta_{\text{peft}_i} - \theta_{\text{peft}_0}$                             ▷ Task vector
  5:      **for** $k = 1$ to $d$ **do**                     ▷ $d$ = number of parameters
  6:          **if** $|\tau_{\text{curr}}^{(k)}| \geq |\tau^{(k)}|$ **then**
  7:              $\tau^{(k)} \leftarrow \tau_{\text{curr}}^{(k)}$            ▷ Keep larger magnitude, preserve sign
  8:          **end if**
  9:      **end for**
10:      $\theta_{\text{merged}} \leftarrow \theta_{\text{peft}_0} + \alpha \cdot \tau$                        ▷ Update merged PEFT
11: **end for**

---

also mitigates forgetting, since earlier classifiers remain frozen and are not modified during subsequent training. After $t$ tasks, the classifier consists of a set of task-specific heads $\{\theta_1^{\text{cls}}, \theta_2^{\text{cls}}, \ldots, \theta_t^{\text{cls}}\}$, and inference is performed by evaluating each head separately and concatenating their outputs:

$$h(x) = \text{concat}\big(h(x; \theta_1^{\text{cls}}), \ h(x; \theta_2^{\text{cls}}), \ \ldots, \ h(x; \theta_t^{\text{cls}})\big).$$

### 3.2 INCREMENTAL KNOWLEDGE CONSOLIDATION

Although pre-trained models possess strong generalization, they still lack the domain-specific knowledge needed to serve as effective feature extractors. As a result, a fine-tuning stage is required. To prevent forgetting during continuous fine-tuning, early methods often assume that task distributions remain close and restrict adaptation to the first task. However, as the number of tasks grows and their distributions diverge, performance on earlier tasks inevitably deteriorates. Inspired by research on model merging, we propose an incremental integration scheme that unifies task-specific backbones into a single consolidated backbone.

We finetune the model—consisting of the pretrained backbone, the PEFT module, and the task classifier—parameterized by $\theta_{\text{pretrained}}, \theta_{\text{peft}_i}, \theta_{\text{cls}_i}$ on task dataset $\mathcal{D}_i$. To limit drift across tasks, each task is initialized from the most recent PEFT parameters $\theta_{\text{peft}_{i-1}}$ rather than the original base, keeping successive solutions close in parameter space—a property shown to be important for stable merging (Li et al., 2025). The finetuning step `finetune` uses SGD with cross-entropy loss.

After finetuning task $i$, we flatten the PEFT parameters into a vector $\tau_{\text{peft}_i}$. Assuming that final parameters should remain close to the pretrained initialization, we subtract the base vector $\tau_{\text{peft}_0}$ and select elements based on the largest absolute deviations. To avoid parameter growth, we retain only the previously merged vector and the current task vector during each merging step. The resulting update vector $\tau$ is added back to the base parameters to obtain the merged PEFT module for task $i$, as summarized in Algorithm 1.

We further include an ablation in Appendix F, comparing Min, Max, and MaxAbs selection rules. All operators yield stable and competitive performance when applied to PEFT modules, reinforcing that merging only PEFT parameters is robust. To our knowledge, no prior work performs parameter-value-based merging without any trimming phase while still achieving effective results.

### 3.3 LOCAL CLASSIFIER ALIGNMENT

In contrast with a generalized feature extractor, task-specific classifiers should remain well separated to achieve strong classification performance. Because datasets from previous tasks are unavailable, updating earlier classifiers during new training would move them away from their previously optimized values. The standard pipeline therefore trains and merges the backbone while freezing the old classifiers, which introduces a mismatch between the unified backbone and those fixed heads. We address this discrepancy with an alignment step that bridges the two components.

Consider the CIL approach where we use a pretrained model such as VIT to produce high-quality embeddings of the input samples, a Gaussian distribution ($\mathcal{N}$ or prototype) to represent a class, and a classifier. This approach has been investigated heavily and often high-performing CIL methods. At each time step $t$, one can use a learning method to train a classifier $h_t$ based on the classifier $h_{t-1}$ which already fitted for prior tasks and a dataset $\boldsymbol{D}_t$ for the current task. The overall performance of $h_t$ depends heavily on the employed learning algorithm. A good algorithm can well align the prototypes and classifier. However, this might not be always the case, and hence can degrade the overall CIL performance.

We propose a simple finetuning step, called *Local Classifier Alignment (LCA)*, to better align the classifier and prototypes. Specifically, LCA minimizes the following loss

$$L(\boldsymbol{D}, h_t) = \frac{1}{C_t} \sum_{i=1}^{C_t} L_i \tag{1}$$

$$L_i = \mathbb{E}_{\boldsymbol{z} \sim \boldsymbol{D}_i} \left[ \ell(h_t, \boldsymbol{z}) \right] + \lambda \mathbb{E}_{\boldsymbol{z}, \boldsymbol{z}' \sim \boldsymbol{D}_i} \left[ \left| \ell(h_t, \boldsymbol{z}) - \ell(h_t, \boldsymbol{z}') \right| \right] \tag{2}$$

where $\boldsymbol{D}_i$ contains i.i.d. samples from the Gaussian distribution $\mathcal{N}_i$ that represents class $i$ in the feature space induced by the backbone, $\boldsymbol{D} = \{\boldsymbol{D}_1, ..., \boldsymbol{D}_t\}$, and $C_t$ is the total number of observed classes upto task $t$.

Basically, LCA tries to simultaneously minimize the loss for each class and keep the loss less sensitive to a small change in the input samples around the class prototypes. The class loss can be seen from the first term $\mathbb{E}_{\boldsymbol{z} \sim \boldsymbol{D}_i}[\ell(h_t, \boldsymbol{z})]$, while the sensitivity comes from the second term in each $L_i$. Such a term can be seen as a regularizer to penalize the classifier for unstable predictions. A larger value for $\lambda$ suggests a stronger penalty for sensitivity of the loss.

It is worth noting that the novelty of LCA comes from the regularization term. It not only helps improve robustness of the classifier, but also can reduce overlapping between classes. Indeed, when using the first term as the main objective for training the classifier, some samples randomly generated by $\mathcal{N}_i$ of one class can lie far from the $i$-th class prototype and hence can be closer to some other class prototypes. This can harm the training for the classifier. The second term can help us reduce the negative effect from those potentially harmful samples, under a suitable choice of $\lambda$.

### 3.4 THEORETICAL ANALYSIS

We next analyze the generalization ability of the classifier $h_t$. Until time step $t$, the classifier $h_t$ already has learned from $C_t$ classes. We want to estimate its test error for those learned tasks. To this end, the classical tradition is to estimate $L(P, h_t) = \frac{1}{C_t} \sum_{i=1}^{C_t} \mathbb{E}_{\boldsymbol{z} \sim \mathcal{N}_i} [\ell(h_t, \boldsymbol{z})]$, which represents the overall expected error for the learned tasks.

Let $\bigcup_{i=1}^t \mathcal{Z}_i$ be the decomposition of the data space into non-overlapping local areas, with $\mathcal{Z}_i$ as the local area with centroid $\mu_i$, where $\mu_i$ is the mean of the Gaussian distribution $\mathcal{N}_i$, for each index $i$. We have the following bound for the expected error of the overall classifier up to time step $t$, whose proof appears in Appendix A.

**Theorem 3.1.** *Consider a model $h_t$ learned from a dataset $\boldsymbol{D} = \{\boldsymbol{D}_1, ..., \boldsymbol{D}_t\}$, where $\boldsymbol{D}_i$ contains $n_i$ i.i.d. samples from distribution $\mathcal{N}_i$ for each $i \leq C_t$, and a bounded loss $\ell$. Denote $P = \frac{1}{C_t} \sum_{i=1}^{C_t} \mathcal{N}_i$ as the overall distribution, $n = \sum_{i=1}^{C_t} n_i$, $\ell_{\max} = \sup_{\boldsymbol{z}} \ell(h_t, \boldsymbol{z})$, and $\bar{\epsilon}_i(h_t) = \mathbb{E}_{\boldsymbol{z} \sim \mathcal{N}_i, \boldsymbol{s} \sim \boldsymbol{D}_i}[|\ell(h_t, \boldsymbol{z}) - \ell(h_t, \boldsymbol{s})| : \boldsymbol{z}, \boldsymbol{s} \in \mathcal{Z}_i]$ for each index $i$. For any $\delta > 0$, the following holds with probability at least $1 - \delta$:*

$$L(P, h_t) \leq L(\boldsymbol{D}, h_t) + \sum_{i=1}^{C_t} \frac{n_i}{n} \bar{\epsilon}_i(h_t) + \ell_{\max} \sqrt{\frac{C_t \ln 4 + 2 \ln(1/\delta)}{n}} \tag{3}$$

This result shows that the test error of the classifier $h_t$ can be controlled by both the training error and the robustness term $\bar{\epsilon} = \sum_{i=1}^{C_t} \frac{n_i}{n} \bar{\epsilon}_i(h_t)$, which tells how robust is the loss w.r.t. to a small change in the input of the samples around the class prototypes. A stronger robustness (i.e., smaller $\bar{\epsilon}$) can lead to a tighter bound on the test error, suggesting a better model. When both $L(\boldsymbol{D}, h_t)$ and $\bar{\epsilon}$ are small, the model must have small test error and hence generalize well on unseen data. On the other hand, a bad model will exhibit a large training error $L(\boldsymbol{D}, h_t)$ or large $\bar{\epsilon}$. Therefore, Theorem 3.1 plays as

the theoretical foundation for our LCA loss (1). Training by this loss would arguably improve both performance and robustness of the classifier, which is crucial for real-world CIL tasks.

**Corollary 1.** *Given the notations and assumptions as in Theorem 3.1, if $n_i = m$ for all $i$, then the following holds with probability at least $1 - \delta$:*

$$L(P, h_t) \leq L(\boldsymbol{D}, h_t) + \frac{1}{C_t} \sum_{i=1}^{C_t} \bar{\epsilon}_i(h_t) + \ell_{\max} \sqrt{\frac{\ln 4}{m} + \frac{2 \ln(1/\delta)}{mC_t}} \tag{4}$$

This is a direct consequence of Theorem 3.1. It suggests that, to assure high generalization, the number $m$ of samples for each class should not be too small when doing alignment. If one use a small $m$, the uncertainty part in (4) will be large, meaning some randomness (by noise) can harm the alignment step.

*Remark* 1. Although the LCA loss (1) is introduced to the CIL context, the loss is general enough to be employed in many other contexts. Indeed, one can use LCA loss to train a CIL classifier. Also, one can easily plug LCA to do a finetuning step for the existing CIL methods. In those cases, the theoretical benefits of LCA shown in Theorem 3.1 may remain valid. It is worth noting that the result in Theorem 3.1 holds when the feature distributions (or prototypes) are fixed. This means Theorem 3.1 may not directly apply for the cases of prototype changes.

To fully understand the generalization dynamic of the overall CIL classifier, the previous results are insufficient. We need to take the backbone change into consideration. A change to the main backbone will lead to a change in the class prototypes that represent the Gaussian $\mathcal{N}_i$. As a result, the overall distribution $P$ in Theorem 3.1 will change after learning a new task.

After training task $t - 1$, $\mathcal{N}_i$ can accurately represent each new class $i$ of task $t - 1$ and $P_{t-1} = \frac{1}{C_{t-1}} \sum_{i=1}^{C_{t-1}} \mathcal{N}_i$ represents the overall distribution. After training a new task $t$, each distribution $\mathcal{N}_i$ may change to $\widehat{\mathcal{N}}_i$ due to backbone change. Therefore, the overall distribution until task $t$ can be rewritten as $\widehat{P}_t = \frac{1}{C_t} \sum_{i=1}^{C_{t-1}} \widehat{\mathcal{N}}_i + \frac{1}{C_t} \sum_{j=C_{t-1}}^{C_t} \mathcal{N}_j$. However using this distribution to analyze generalization ability for all trained tasks may not be accurate. What we should analyze is for the distribution $P_t = \frac{1}{C_t} \sum_{i=1}^{C_t} \mathcal{N}_i$. We have the following results.

**Theorem 3.2.** *Given the notations in Theorem 3.1, we have*

$$L(P_t, h_t) \leq 2\ell_{\max} \mathsf{TV}(P_t, \widehat{P}_t) + L(\widehat{P}_t, h_t) \tag{5}$$

*where $\mathsf{TV}(\cdot, \cdot)$ denotes the total variation distance. Let $\hat{\boldsymbol{D}} = \bigcup_{i \leq C_{t-1}} \hat{\boldsymbol{D}}_i \bigcup_{j=C_{t-1}}^{C_t} \boldsymbol{D}_j$, with $\hat{\boldsymbol{D}}_i \sim \widehat{\mathcal{N}}_i$ and $\boldsymbol{D}_j \sim \mathcal{N}_j$, contains i.i.d. samples. For any $\delta > 0$, the following holds with probability at least $1 - \delta$:*

$$L(P_t, h_t) \leq 2\ell_{\max} \mathsf{TV}(P_t, \widehat{P}_t) + L(\hat{\boldsymbol{D}}, h_t) + \sum_{i=1}^{C_t} \frac{n_i}{n} \bar{\epsilon}_i(h_t) + \ell_{\max} \sqrt{\frac{C_t \ln 4 + 2 \ln(1/\delta)}{n}} \tag{6}$$

Basically, this theorem says that the test error of model $h_t$ will be small when one can ensure that the LCA loss ($L(\hat{\boldsymbol{D}}, h_t) + \sum_{i=1}^{C_t} \frac{n_i}{n} \bar{\epsilon}_i(h_t)$) is small and the induced distribution $\widehat{P}_t$ is close to $P_t$ (the accurate feature distribution). When distributions $\widehat{P}_t$ and $P_t$ are far from each other, the backbone really causes catastrophic forgetting and thus can significantly increase the errors for past tasks. As a result, bound (6) suggests that a good CIL model should not significantly change the feature distributions of past tasks while being both robust and accurate for each class.

Return to our CIL method, both components have their own important roles. IM can train the backbone to adapt the feature distribution to the new task, but also reduces catastrophic forgetting for past tasks and hence keeps small $\mathsf{TV}(P_t, \widehat{P}_t)$. The classifier alignment by LCA can simultaneously train the new part, adapt the old part while encouraging robustness of the overall classifier. Those roles are very crucial to ensure high performance for CIL. Therefore those analyses reveal the foundational support for LCA.

Table 1: Average performance comparison on seven datasets with ViT-B/16-IN1K as the pretrained backbone. The best result is highlighted in bold, while the second best is highlighted in italic.

| Method | CIFAR100 | IN-R | IN-A | CUB | OB | VTAB | CARS | Overall |
|---|---|---|---|---|---|---|---|---|
| APER+Adapter | 90.8 ± 0.5 | 78.8 ± 0.6 | 58.9 ± 1.3 | 89.7 ± 1.3 | 80.3 ± 0.4 | 90.7 ± 0.6 | 50.6 ± 1.1 | 77.1 |
| APER+Finetune | 81.7 ± 0.9 | 72.1 ± 0.8 | 58.7 ± 3.7 | 89.5 ± 1.4 | 77.8 ± 1.2 | 91.8 ± 1.4 | 53.2 ± 1.4 | 75.0 |
| APER+SSF | 89.5 ± 1.0 | 78.1 ± 1.1 | 61.6 ± 0.5 | 89.6 ± 1.1 | 80.3 ± 0.6 | 91.8 ± 1.5 | 51.3 ± 1.1 | 77.5 |
| APER+VPT-Deep | 89.0 ± 0.9 | 78.8 ± 0.7 | 57.0 ± 0.4 | 89.0 ± 1.2 | 79.8 ± 0.4 | 91.9 ± 1.4 | 50.6 ± 3.0 | 76.6 |
| APER+VPT-Shallow | 88.1 ± 0.9 | 67.3 ± 3.5 | 56.9 ± 1.4 | 89.5 ± 1.4 | 79.7 ± 0.9 | 91.5 ± 0.8 | 50.9 ± 0.8 | 74.8 |
| CODA-Prompt | 91.0 ± 0.2 | 78.2 ± 0.4 | 48.1 ± 0.9 | 75.6 ± 1.2 | 71.0 ± 0.1 | 65.6 ± 2.6 | 26.3 ± 0.6 | 65.1 |
| DualPrompt | 86.7 ± 0.6 | 74.6 ± 0.5 | 55.3 ± 1.5 | 78.9 ± 1.0 | 74.4 ± 1.2 | 84.0 ± 5.9 | 49.4 ± 2.1 | 71.9 |
| EASE | 91.7 ± 0.3 | 82.4 ± 0.5 | *67.8 ± 1.8* | 89.5 ± 1.2 | 80.8 ± 0.2 | *93.3 ± 0.1* | 48.1 ± 1.2 | 79.1 |
| L2P | 87.7 ± 1.4 | 77.3 ± 0.6 | 52.6 ± 1.7 | 75.8 ± 1.8 | 73.8 ± 1.2 | 82.4 ± 2.8 | 53.4 ± 1.2 | 71.9 |
| MOS | *94.3 ± 0.3* | 83.3 ± 0.6 | 67.6 ± 2.0 | **92.3 ± 0.6** | **86.1 ± 0.7** | 92.4 ± 0.5 | 71.4 ± 19.6 | *83.9* |
| SLCA | 93.7 ± 0.3 | *85.1 ± 0.3* | 45.1 ± 19.8 | 90.2 ± 0.9 | *82.7 ± 0.6* | 91.1 ± 3.4 | 74.6 ± 2.2 | 80.4 |
| IM | 92.8 ± 0.1 | 84.3 ± 1.0 | 66.5 ± 1.1 | 86.7 ± 0.8 | 81.1 ± 0.8 | 84.6 ± 4.9 | 70.1 ± 1.5 | 80.9 |
| IM+LCA | **94.8 ± 0.3** | **85.8 ± 0.2** | **75.0 ± 0.5** | *90.8 ± 0.3* | 81.4 ± 0.5 | **95.2 ± 1.1** | **76.2 ± 1.4** | **85.6** |

# 4 EXPERIMENTS

This section presents the details of our experiments on seven benchmark datasets, along with an ablation study to examine some design aspects of our method.

## 4.1 EXPERIMENT SETUP

**Dataset.** We follow the setup in (McDonnell et al., 2023) to select datasets and define the number of classes in each incremental task. Specifically, we evaluate on CIFAR100 (Krizhevsky & Hinton, 2009), ImageNet-R (IN-R) (Hendrycks et al., 2021a), ImageNet-A (IN-A) (Hendrycks et al., 2021b), CUB-200 (CUB) (Welinder et al., 2010), OmniBenchmark (OB) (Zhang et al., 2022), VTAB (Zhai et al., 2019), and StanfordCars (CARS) (Krause et al., 2013). All datasets are split into 10 tasks, except VTAB which is divided into 5.

**Baselines.** the following methods are used for comparison:

- **Ours:** We consider two variants: **IM** (Incremental Merging) and **IM+LCA**, where the latter incorporates LCA in the alignment phase.
- Pre-trained based CIL methods: *CODA-Prompt* (Smith et al., 2023), *DualPrompt* (Wang et al., 2022b), *L2P* (Wang et al., 2022c), *EASE* (Zhou et al., 2024), *MOS* (Sun et al., 2025b), *SLCA* (Zhang et al., 2023), *APER* (Zhou et al., 2025).

For fair comparison and reproducibility, we adopt the implementation from (Sun et al., 2025a) for all baseline methods and use the same pre-trained backbone, ViT-B/16 trained on ImageNet-1K (ViT-B/16-IN1K) (Dosovitskiy et al., 2020).

**Training.** We apply LoRA (Hu et al., 2022) and ensure a consistent computational budget by fixing the low-rank dimension to 64. All experiments are carried out on a single NVIDIA RTX 4090 GPU running Ubuntu 22.04. Detail on hyperparameters is mentioned in Appendix B.

**Evaluation Metrics.** Following standard practice, we report the performance along incremental stages, defined as $\text{AA} = \frac{1}{T} \sum_{i=1}^{T} A_t$, where $A_t$ denotes the average accuracy after done training on task $t$.

## 4.2 EXPERIMENT RESULTS

### 4.2.1 OVERALL BENCHMARK

Table 1 reports the mean and standard deviation of accuracy across three random seeds (1993, 1994, 1995). With LCA, the final model achieves the highest performance on five out of seven benchmark datasets, yielding an overall improvement of nearly 2%. Unlike methods such as EASE (Zhou et al., 2024) and MOS (Sun et al., 2025b), which either expand the backbone to integrate new tasks or rely on complex inference procedures, our approach requires no additional mechanism. The backbone is

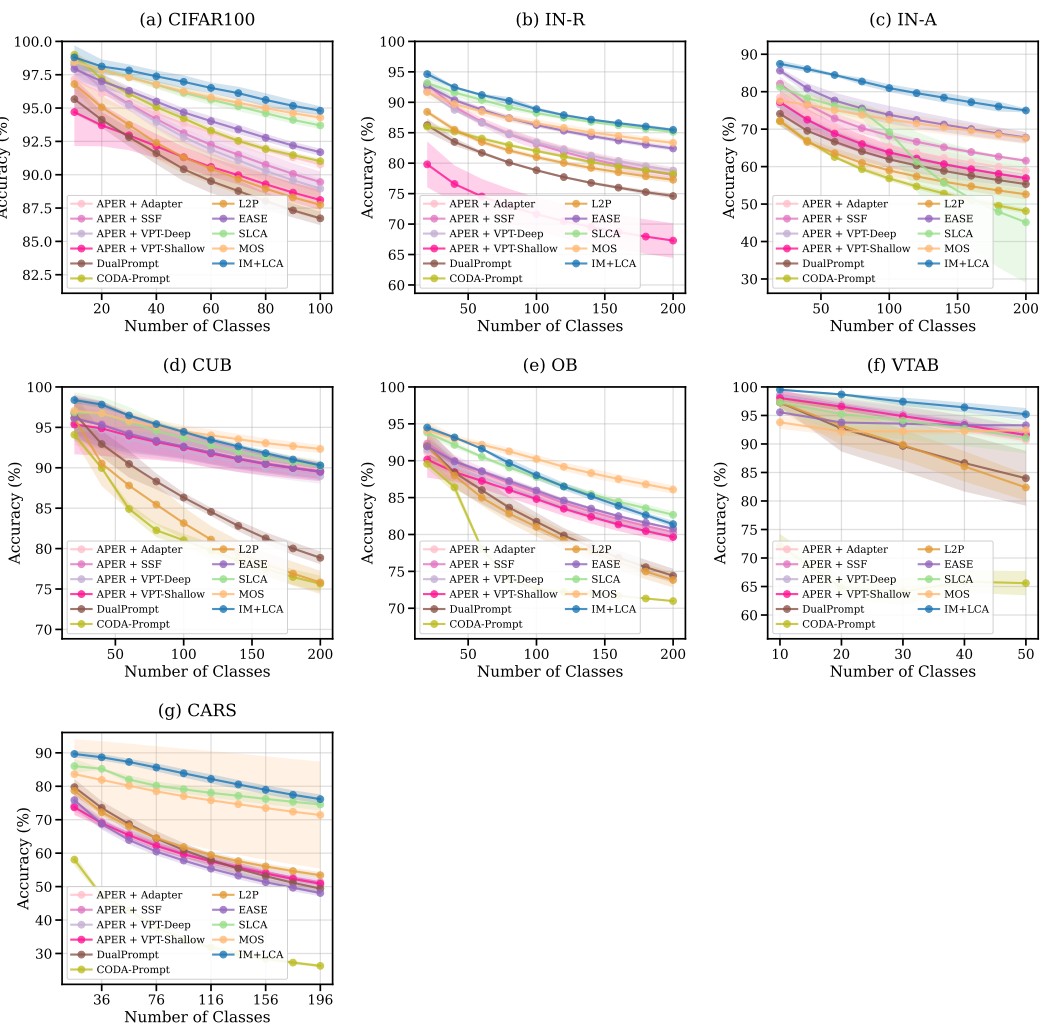

Figure 2: Performance curves of different methods across all tasks and datasets. All methods use ViT-B/16-IN1K as the pre-trained backbone without any additional exemplars.

incrementally merged without storing past parameters or samples, and the only extra storage is the mean and covariance of each encountered class, which scales as $\mathcal{O}(n)$ with the number of classes. Figure 2 further shows that LCA consistently outperforms other methods across most datasets, with a notable improvement of 8% on ImageNet-A compared to the runner-up.

### 4.2.2 ROBUSTNESS MEASUREMENT

Following the standard (Hendrycks & Dietterich, 2019; Taori et al., 2020), we measure the performance of IM and IM+LCA on two robustness benchmarks, CIFAR100-C and CIFAR100-P. CIFAR100-C is constructed by applying 19 types of common corruptions (e.g., noise, blur, weather, and digital distortions) at 5 severity levels to the original CIFAR-100 test set, thereby evaluating model robustness under distribution shift. The final corruption robustness accuracy is computed as

$$\text{Acc}_{\text{C}} = \frac{1}{19 \times 5} \sum_{c=1}^{19} \sum_{s=1}^{5} A_{c,s},$$

where $A_{c,s}$ is the accuracy under corruption type $c$ with severity $s$.

CIFAR100-P, in contrast, focuses on prediction stability under perturbations. Each image is perturbed by transformations such as translations, rotations, or noise, and the model's consistency is

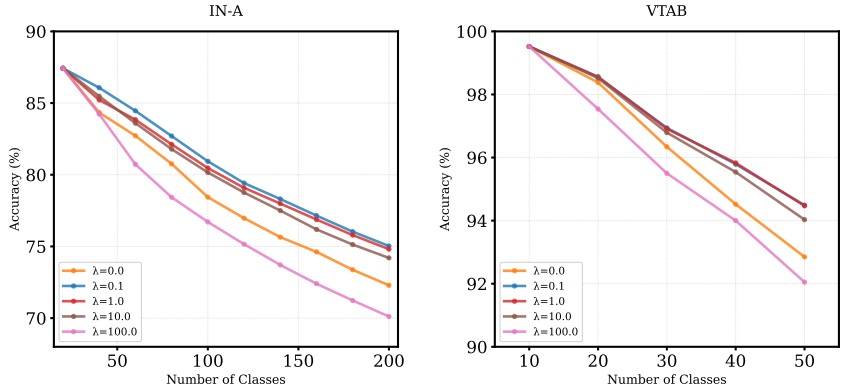

Figure 3: Effect of $\lambda$ on the accuracy on two datasets.

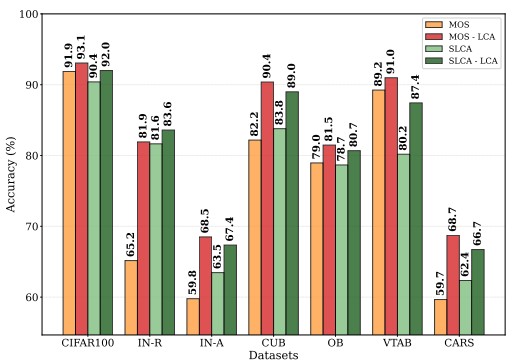

(a) Performance for variants of MOS and SLCA. The postfix LCA means that method uses our alignment loss.

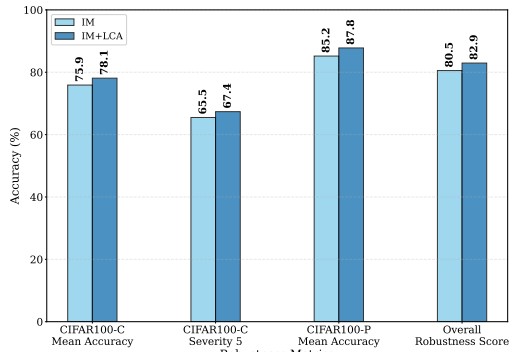

(b) Robustness comparison between IM and IM+LCA on CIFAR100-C and CIFAR100-P. Metrics include mean accuracy, accuracy at severity level 5, and overall robustness score.

Figure 4: (a) Complementary evaluation of LCA when using LCA for MOS and SLCA. (b) Robustness performance of IM and IM+LCA on corruption and perturbation benchmarks.

measured across perturbed versions. The perturbation robustness accuracy is defined as

$$\text{Acc}_{\text{P}} = \frac{1}{|\mathcal{P}|} \sum_{p \in \mathcal{P}} \frac{1}{N_p} \sum_{i=1}^{N_p} A_{p,i},$$

where $\mathcal{P}$ denotes the set of perturbation types, $N_p$ is the number of perturbed samples for type $p$, and $A_{p,i}$ is the accuracy on the $i$-th perturbed sample. Finally, to summarize robustness across both benchmarks, we report an overall robustness score:

$$\text{Robustness} = \tfrac{1}{2} \left( \text{Acc}_{\text{C}} + \text{Acc}_{\text{P}} \right).$$

Together, these benchmarks evaluate both accuracy under distributional corruption and resilience of predictions under small perturbations.

Figures 4b and 5 summarize the robustness results. Figure 4b shows that when being trained with LCA, the model obtains a clear improvement on robustness, with more than +2% gain in mean accuracy on CIFAR100-C and a +2.5% gain on CIFAR100-P. The radar plots in Figure 5 further reveal that LCA consistently improves accuracy across all corruption and perturbation types. These results confirm that LCA strengthens robustness both on average and across diverse perturbations.

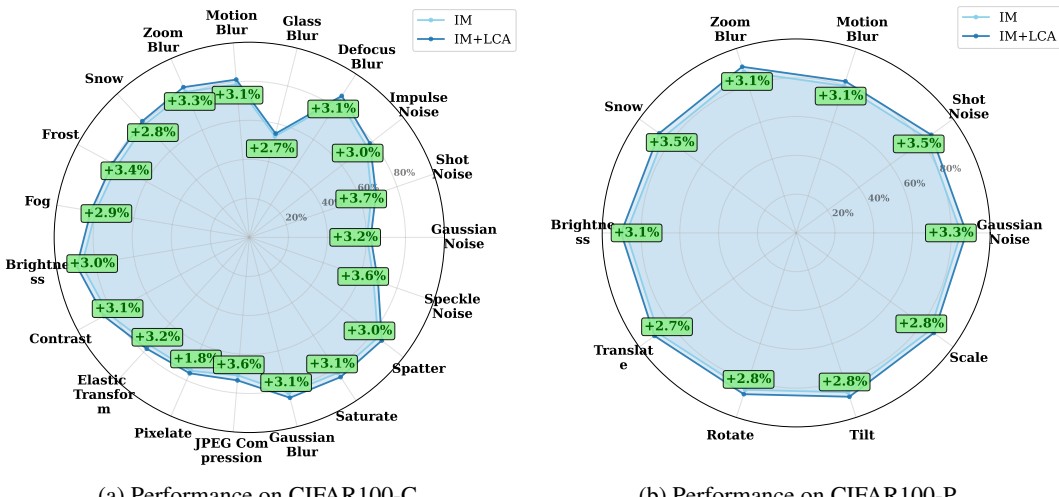

(a) Performance on CIFAR100-C.
(b) Performance on CIFAR100-P.

Figure 5: Accuracy performance of IM and IM+LCA under different corruption and perturbation types. The relative difference between IM and IM+LCA is highlighted.

### 4.2.3 ABLATION STUDY

**LCA as a complementary component.** We further evaluate the applicability of LCA on SLCA (Zhang et al., 2023) and MOS (Sun et al., 2025b), two methods that also update the backbone progressively. To asses the impact of our proposed method, we construct two baselines by omitting the final step of these methods, and retaining only the update backbone part. From these, we build their counterparts, SLCA-LCA and MOS-LCA, which include the aligment step as the final step. We measure the average performance and report in Figure 4a. We do not find the optimal hyperparameters but fixing the value of $\lambda$ at 0.1, yet the variants with LCA show improvements on all scenerios, notably in IN-A, CUB, VTAB, and CARS, in some cases even matching the reported results of the original methods (SLCA-LCA achieves 89.0% in CUB compare to 90.0% of the original, MOS-LCA achieves 93.1% in CIFAR100 compare to 94.3%).

**Hyperparameter sensitivity.** In our method, $\lambda$ controls the strength of the robustness penalty in the loss function. While an appropriate choice of $\lambda$ is important for achieving strong performance as Figure 3 shows that overly strong regularization can degrade results. In practice, we find that $\lambda = 0.1$ provides stable and reliable performance across all datasets.

## 5 CONCLUSION

This paper investigates the mismatch that may arise between a continuously updated backbone and the task-specific classifiers in continual learning. We propose a theoretically grounded loss function, with a provable bound on classification error, which enables the training of all classifiers using features generated from a Gaussian distribution. Extensive experiments on seven benchmark datasets confirm the effectiveness of our method, and additional robustness evaluations under various noisy conditions demonstrate consistent improvements. While this work primarily focuses on addressing the alignment phase, future research will explore integrating the proposed loss into the end-to-end training pipeline. Such an approach has the potential to further enhance the robustness of the backbone itself and the performance of the overall CIL method.

Despite having significant contributions, our work still remains some limitations. For example, we have not investigated the proposed LCA loss in other contexts. Furthermore, although providing a theoretical foundation and novel insights, the developed theory does not take the training of the whole backbone into consideration. Addressing those limitations can open interesting directions for future research.

## REPRODUCIBILITY STATEMENT

We have taken several steps to ensure the reproducibility of our results. The hyperparameters required for both training and inference are explicitly reported in Appendix B. The complete source code is provided as supplementary material, where users can download and unzip the package, follow the installation instructions, and run the main script to reproduce our experiments. All datasets used in this work are publicly available, implementation details and evaluation protocols are described in Section 4.1. Together, these resources are intended to make it straightforward for others to replicate and build upon our work.

## ACKNOWLEDGEMENTS

This work was supported by JSPS Grant-in-Aid for Challenging Exploratory Research - Grant Number JP25K22833 and JSPS Research on Academic Transformation Areas (A) - Grant Number JP22H05194. The authors also gratefully acknowledge the financial support provided by Hattori Hokokai Foundation and Telecommunications Advancement Foundation.

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

# A  PROOF FOR MAIN THEOREMS

*Proof of Theorem 3.1.* Let $p_i = P(\mathcal{Z}_i)$ be the probability measure of area $\mathcal{Z}_i$, and note that $\sum_{i=1}^{t} p_i = 1$.

We first observe that

$$L(P, h_t) = L(P, h_t) - \sum_{i=1}^{C_t} \frac{n_i}{n} L(\mathcal{N}_i, h_t) + \sum_{i=1}^{C_t} \frac{n_i}{n} L(\mathcal{N}_i, h_t) - L(\mathbf{D}, h_t) + L(\mathbf{D}, h_t) \quad (7)$$

Since $L(P, h_t) = \sum_{i=1}^{C_t} p_i L(\mathcal{N}_i, h_t)$ and $L(\mathcal{N}_i, h_t) \leq \ell_{\max}$, we observe that

$$L(P, h_t) - \sum_{i=1}^{C_t} \frac{n_i}{n} L(\mathcal{N}_i, h_t) = \sum_{i=1}^{C_t} p_i L(\mathcal{N}_i, h_t) - \sum_{i=1}^{C_t} \frac{n_i}{n} L(\mathcal{N}_i, h_t) \quad (8)$$

$$= \sum_{i=1}^{C_t} \left( p_i - \frac{n_i}{n} \right) L(\mathcal{N}_i, h_t) \quad (9)$$

$$\leq \ell_{\max} \sum_{i=1}^{C_t} \left| p_i - \frac{n_i}{n} \right| \quad (10)$$

Note that $(n_1, ..., n_{C_t})$ is a multinomial random variable with parameters $n$ and $(p_1, ..., p_{C_t})$. For any $\epsilon > 0$, Bretagnolle-Huber-Carol inequality shows $\Pr\left( \sum_{i=1}^{C_t} \left| p_i - \frac{n_i}{n} \right| \geq 2\epsilon \right) \leq 2^{C_t} \exp(-2n\epsilon^2)$. In other words, for any $\delta > 0$, taking $\epsilon = \sqrt{\frac{C_t \ln 2 - \ln \delta}{2n}}$, the following holds true with probability at least $1 - \delta$:

$$L(P, h_t) - \sum_{i=1}^{C_t} \frac{n_i}{n} L(\mathcal{N}_i, h_t) \leq C\sqrt{\frac{C_t \ln 4 - 2 \ln \delta}{n}} \quad (11)$$

Next we observe the second term:

$$\sum_{i=1}^{C_t} \frac{n_i}{n} L(\mathcal{N}_i, h_t) - L(\mathbf{D}, h_t) = \sum_{i=1}^{C_t} \frac{n_i}{n} L(\mathcal{N}_i, h_t) - \sum_{i=1}^{C_t} \frac{n_i}{n} L(\mathbf{D}_i, h_t) \quad (12)$$

$$= \sum_{i=1}^{C_t} \frac{n_i}{n} \left[ L(\mathcal{N}_i, h_t) - L(\mathbf{D}_i, h_t) \right] \quad (13)$$

$$= \sum_{i=1}^{C_t} \frac{n_i}{n} \left( \mathbb{E}_{\mathbf{z} \sim \mathcal{N}_i} [\ell(h_t, \mathbf{z}) - L(\mathbf{D}_i, h_t) : \mathbf{z} \in \mathcal{Z}_i] \right) \quad (14)$$

$$= \sum_{i=1}^{C_t} \frac{n_i}{n} \left( \mathbb{E}_{\mathbf{z} \sim \mathcal{N}_i, \mathbf{s} \sim \mathbf{D}_i} [\ell(h_t, \mathbf{z}) - \ell(h_t, \mathbf{s}) : \mathbf{z}, \mathbf{s} \in \mathcal{Z}_i] \right) \quad (15)$$

$$\leq \sum_{i=1}^{C_t} \frac{n_i}{n} \left( \mathbb{E}_{\mathbf{z} \sim \mathcal{N}_i, \mathbf{s} \sim \mathbf{D}_i} [|\ell(h_t, \mathbf{z}) - \ell(h_t, \mathbf{s})| : \mathbf{z}, \mathbf{s} \in \mathcal{Z}_i] \right) \quad (16)$$

$$= \sum_{i=1}^{C_t} \frac{n_i}{n} \bar{\epsilon}_i(h_t) \quad (17)$$

Combining (7) and (11) and (17) completes the proof. $\qquad \square$

*Proof of Theorem 3.2.* Denote $p_t(\boldsymbol{z})$ and $\hat{p}_t(\boldsymbol{z})$ as the density of distributions $P_t$ and $\widehat{P}_t$, respectively. Observe that

$$
\begin{align}
L(P_t, h_t) &= L(P_t, h_t) - L(\widehat{P}_t, h_t) + L(\widehat{P}_t, h_t) \tag{18}\\
&= \mathbb{E}_{\boldsymbol{z} \sim P_t}[\ell(h_t, \boldsymbol{z})] - \mathbb{E}_{\boldsymbol{z} \sim \widehat{P}_t}[\ell(h_t, \boldsymbol{z})] + L(\widehat{P}_t, h_t) \tag{19}\\
&= \int \ell(h_t, \boldsymbol{z}) p_t(\boldsymbol{z}) d\boldsymbol{z} - \int \ell(h_t, \boldsymbol{z}) \hat{p}_t(\boldsymbol{z}) d\boldsymbol{z} + L(\widehat{P}_t, h_t) \tag{20}\\
&\leq \int |\ell(h_t, \boldsymbol{z})| \cdot |p_t(\boldsymbol{z}) d\boldsymbol{z} - \hat{p}_t(\boldsymbol{z})| d\boldsymbol{z} + L(\widehat{P}_t, h_t) \tag{21}\\
&\leq \ell_{\max} \int |p_t(\boldsymbol{z}) d\boldsymbol{z} - \hat{p}_t(\boldsymbol{z})| d\boldsymbol{z} + L(\widehat{P}_t, h_t) \tag{22}\\
&= 2\ell_{\max} \mathsf{TV}(P_t, \widehat{P}_t) + L(\widehat{P}_t, h_t) \tag{23}
\end{align}
$$

which shows the first statement. The second result thus follows using the result of Theorem 3.1. $\square$

## B  TRAINING DETAILS

Table 2: Training and merging hyperparameters used in all experiments of LCA.

| Hyperparameter | Value |
|---|---|
| Training epochs | 10 |
| Batch size | 64 |
| Base learning rate | $1 \times 10^{-2}$ |
| Weight decay | $5 \times 10^{-4}$ |
| Optimizer | SGD (momentum = 0.9) |
| Learning rate scheduler | CosineAnnealing (eta-min = $1 \times 10^{-6}$) |
| Merging coefficient $\alpha$ | 1.0 |
| Alignment classifier epochs | 10 |
| Number of samples per class | 512 |
| Alignment batch size | 128 |
| LoRA rank | 64 |
| LoRA alpha | 128 |
| LoRA dropout | 0.0 |
| LoRA initialization | Gaussian |

## C  DETAILS OF EXAMPLE GENERATION

In this section, we provide details on how Gaussian distributions are used to align the classifiers during the incremental training process. Pre-trained models typically produce well-structured representations, which allows us to approximate each class distribution using its empirical mean and covariance.

**Storing class statistics.** For each class $c$ at training stage $t$, we extract features using the backbone parameters $\theta_t$. The empirical mean $\mu_c \in \mathbb{R}^d$ and covariance $\Sigma_c \in \mathbb{R}^{d \times d}$ are computed as

$$
\mu_c = \frac{1}{K} \sum_{i=1}^{|\mathcal{D}_t|} \mathbf{1}(y_i = c)\, \phi(x_i; \theta_t), \tag{24}
$$

$$
\Sigma_c = \frac{1}{K} \sum_{i=1}^{|\mathcal{D}_t|} \big(\phi(x_i; \theta_t) - \mu_c\big)\big(\phi(x_i; \theta_t) - \mu_c\big)^\top, \tag{25}
$$

where $\phi(\cdot; \theta_t)$ denotes the feature extractor with backbone parameters $\theta_t$, and $K = \sum_{i=1}^{|\mathcal{D}_t|} \mathbf{1}(y_i = c)$.

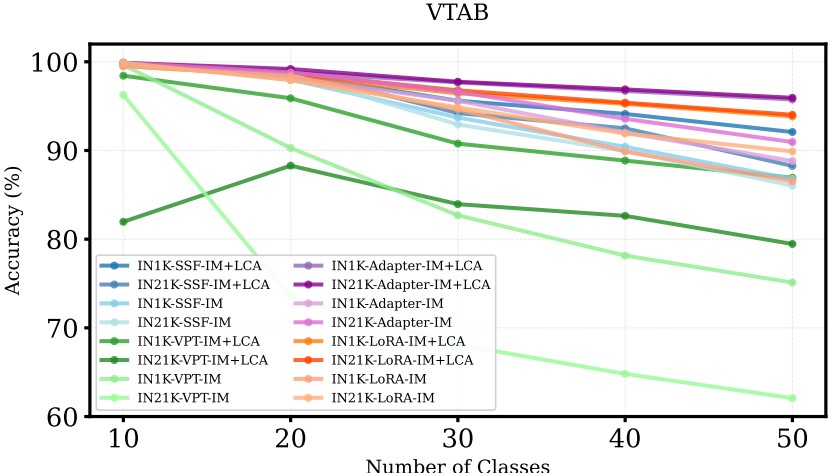

Figure 6: Ablation study on different PEFT strategies with two backbones, ViT-B/16-1K and ViT-B/16-21K.

**Feature replay via Gaussian sampling.** Before each alignment step, we regenerate features for every class $c \in \mathcal{Y}_t$ by sampling from a multivariate Gaussian distribution:

$$\hat{z}_c \sim \mathcal{N}(\mu_c, \Sigma_c). \tag{26}$$

We generate approximately five times the batch size of synthetic features per class, which provides sufficient diversity for classifier alignment.

## D  FURTHER ANALYSIS

**Results with different fine-tuning strategies.** We further examine the adaptability of incremental merging and LCA with different parameter-efficient fine-tuning (PEFT) strategies on the VTAB dataset. The strategies include SSF (Lian et al., 2022), Adapters (Rebuffi et al., 2017), VPT (Jia et al., 2022), and LoRA (Hu et al., 2022), evaluated with ViT-B/16 pretrained on both ImageNet-1K and ImageNet-21K.

Figure 6 shows that LCA consistently improves performance across all PEFT methods. In particular, VPT without LCA suffers a sharp performance drop, suggesting that incremental merging alone is insufficient to prevent forgetting in some cases. Adding LCA, however, proves beneficial across all methods, demonstrating its adaptability. Notably, despite having relatively few trainable parameters, Adapters emerge as a strong candidate within our pipeline, achieving the highest accuracy when combining with LCA.

## E  FURTHER RESULTS ON ACCURACY

In this section, we report additional results of our proposed method when finetuning with the Adapter PEFT module (Houlsby et al., 2019), shown in Figure 7. We refer to this variant as **IM+LCA Adapter**, while the results in the main text correspond to **IM+LCA LoRA**.

From the figure, we observe that several baseline methods suffer from strong instability under certain settings (e.g., SLCA on IN-A and MOS on CARS), whereas our method remains consistently stable across all three random seeds (1993, 1994, 1995). Furthermore, with Adapter, our approach achieves state-of-the-art performance on datasets such as CUB and OB. This confirms that the lower performance reported in the main text for these datasets is due to our reproducibility-oriented constraints on PEFT method choices, rather than limitations of the proposed method itself.

Table 3: Average performance on different merge operators. CA means normal classifier alignment, while LCA is our proposed method.

| Method | IN-A | VTAB | CARS |
|---|---|---|---|
| IM Max | 66.26 ± 0.65 | 84.76 ± 3.69 | 69.90 ± 1.47 |
| IM Max CA | 70.86 ± 1.10 | 92.05 ± 2.44 | 71.88 ± 0.49 |
| IM Max LCA | 73.73 ± 1.13 | 94.22 ± 1.06 | 74.85 ± 1.50 |
| IM Min | 66.49 ± 0.67 | 85.08 ± 3.44 | 69.79 ± 1.36 |
| IM Min CA | 71.11 ± 1.10 | 92.13 ± 2.33 | 71.82 ± 0.50 |
| IM Min LCA | 73.00 ± 1.20 | 94.24 ± 1.11 | 74.28 ± 1.29 |
| IM MaxAbs | 66.5 ± 1.10 | 84.6 ± 4.90 | 70.1 ± 1.50 |
| IM MaxAbs CA | 71.31 ± 0.82 | 91.71 ± 2.24 | 72.15 ± 0.52 |
| IM MaxAbs LCA | 75.0 ± 0.50 | 95.2 ± 1.10 | 76.2 ± 1.40 |

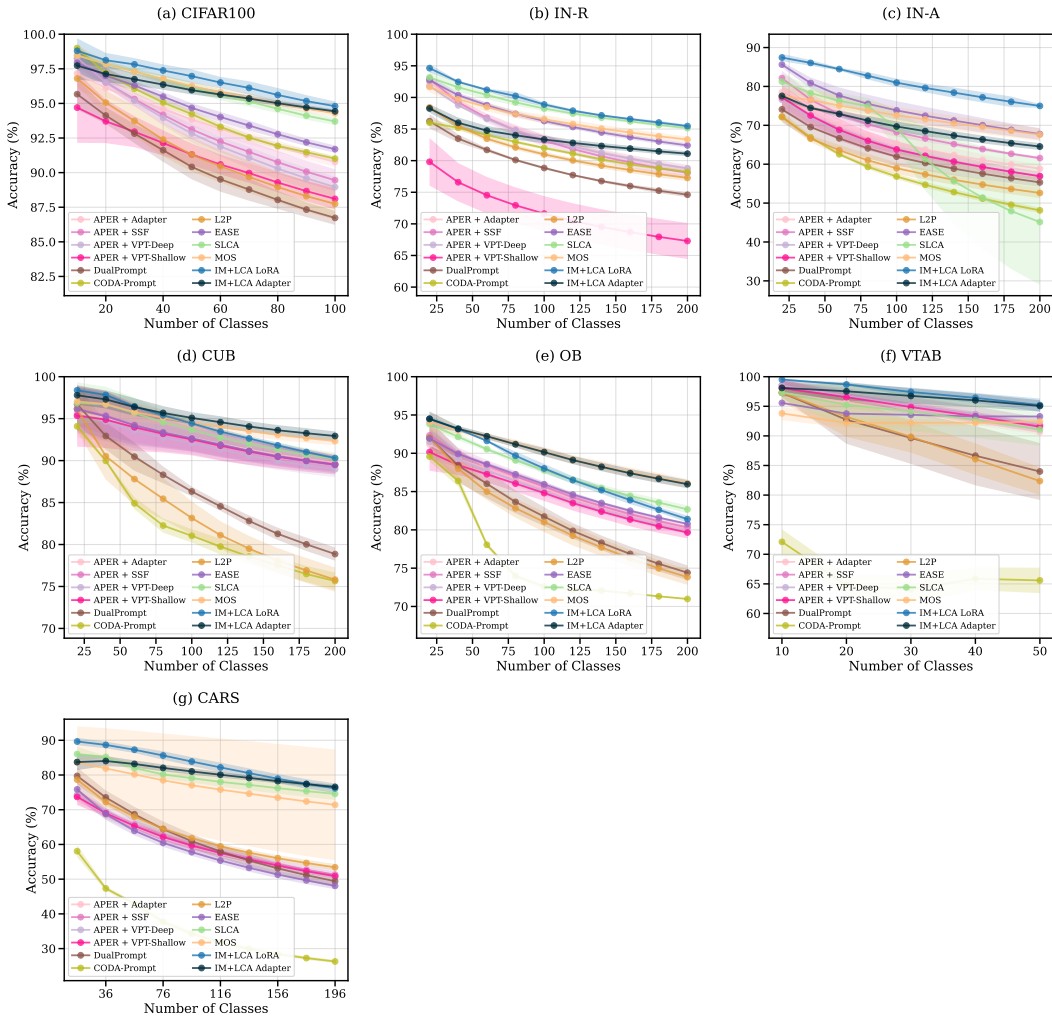

Figure 7: Additional results on Performance Curve.

## F    DIFFERENT MERGING OPERATORS

Table 3 shows the benefits of our parameter-selection design. We evaluate the generalization of our method using three merge operators: Min, which selects the parameter with the smallest value;

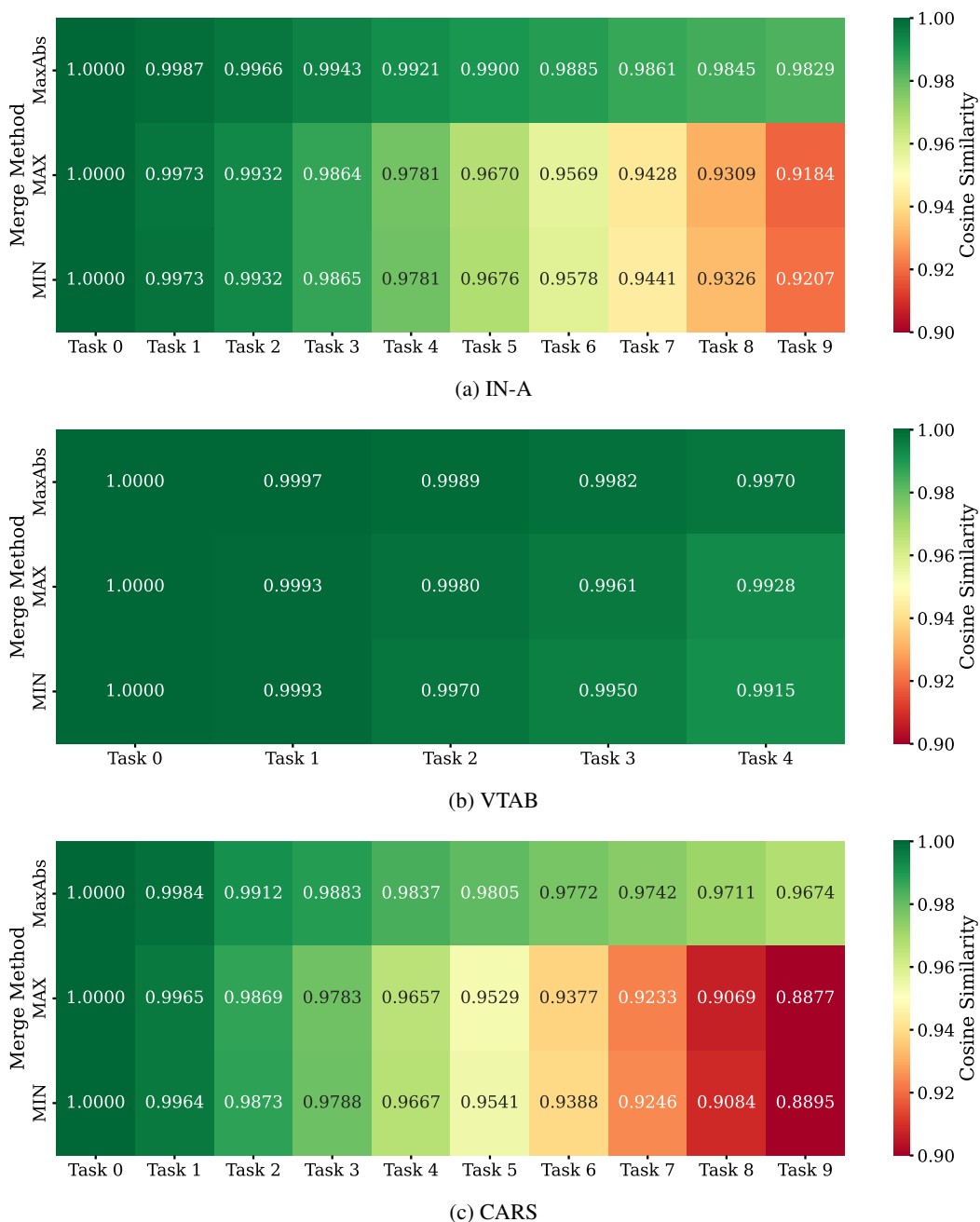

Figure 8: Similarity measurements between backbone representations across tasks, computed relative to the backbone obtained from the first task.

Max, which selects the largest; and MaxAbs, which selects the parameter with the largest absolute magnitude. When we want to reduce the cost of storage by merging only two models, the MaxAbs operator can be viewed as a simplified form of TIES-Merging without the trimming phase in Yadav et al. (2023), and is equivalent to the merge rule proposed in Marczak et al. (2024).

Importantly, unlike prior works such as Marczak et al. (2024), which apply merging to all model parameters, we show that merging only PEFT module parameters is substantially easier and more stable. Under this restricted setting, even very simple operators like Min or Max achieve competitive performance across benchmarks, demonstrating that PEFT-focused merging does not require

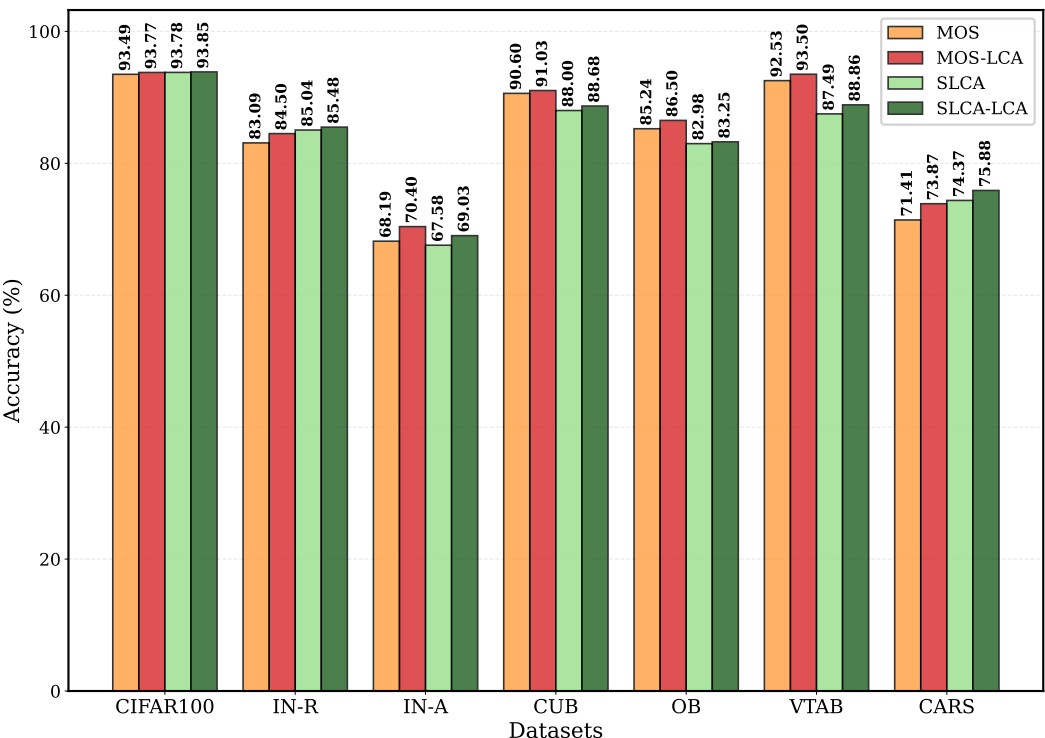

Figure 9: Accuracy of the original methods and their LCA variants across standard datasets.

complex selection strategies. Moreover, we observe that the merge coefficient does not need careful tuning as a fixed value of 1.0 works reliably across all benchmarks.

## G    ABLATION ON ROBUSTNESS TERM

We measure the similarity between the backbone obtained at each task and the backbone learned after the first task. The backbone at the first task serves as an anchor, as it is assumed to bridge the gap between the pre-trained dataset and the target domain. Figure 8 illustrates how incremental fine-tuning with merging enables the backbone to evolve gradually over time, supporting the theoretical intuition discussed in Section 3.4.

Table 3 further shows the impact of including the robustness term during training. Across all merge operators, adding the robustness term yields consistent and clear improvements, demonstrating its effectiveness independent of the chosen merge strategy.

## H    EFFECT OF LCA ON DIFFERENT METHODS

In addition to the results reported in Figure 4a, where we replace the original alignment loss with our LCA loss, we conduct a more extensive ablation to evaluate the adaptability of LCA when attached to other alignment-based methods, specifically SLCA (Zhang et al., 2023) and MOS (Sun et al., 2025b). Both methods continually evolve the backbone representations over time, making them natural candidates for incorporating our proposed loss. Figure 9 shows that, across all datasets, the LCA-augmented variants consistently outperform their corresponding original methods, indicating a clear accuracy improvement.

We further evaluate robustness following the same protocol used for IM. Specifically, we test on CIFAR100-C, the corrupted version of CIFAR100, and CIFAR100-P, the perturbed version. Details of these robustness benchmarks are provided in Section 4.2.2. Figure 10 summarizes the overall

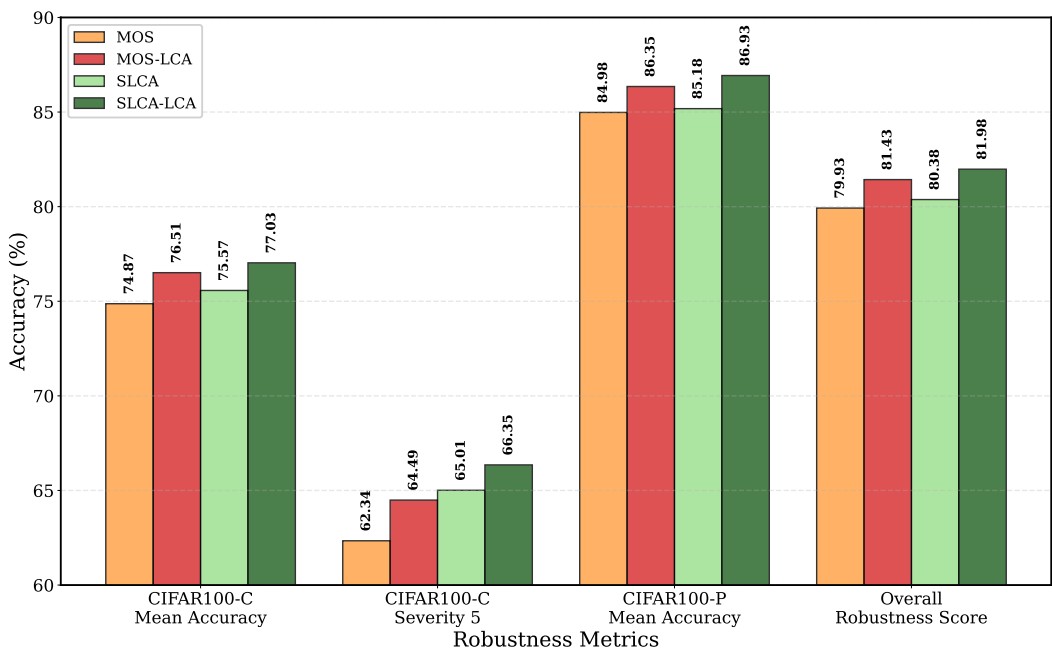

Figure 10: Accuracy of the original methods and their LCA variants under various corruption and perturbation types.

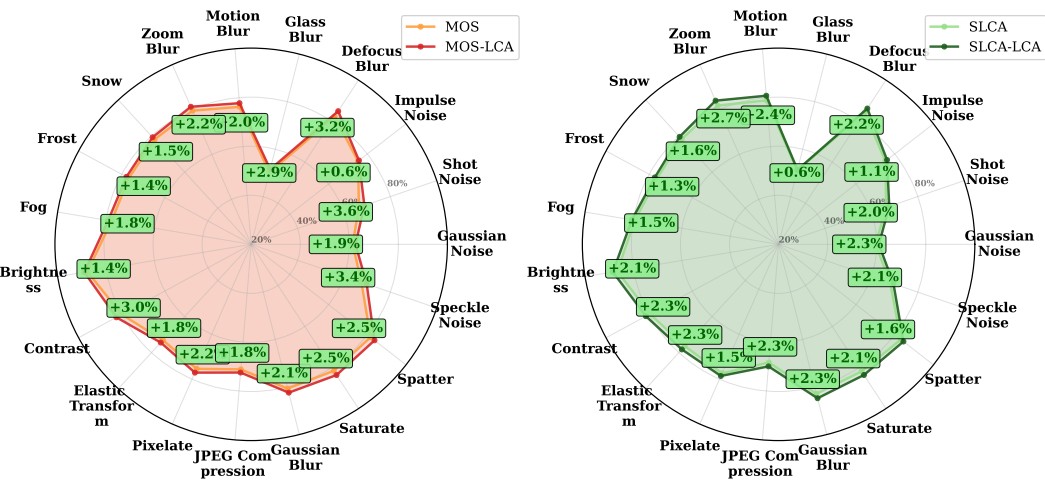

Figure 11: Accuracy of the original methods and their LCA variants on CIFAR100-C corruptions.

robustness gains, while Figures 11 and 12 break down the improvements for each corruption and perturbation type, respectively.

## I    EFFECT OF LCA WHEN SELECTIVELY UPDATING CLASSIFIERS

Updating only half of the task-specific classifier heads (five in IN-A and three in VTAB) during alignment still yields competitive performance as demonstrated in figure 13, indicating that full classifier finetuning is not always necessary.

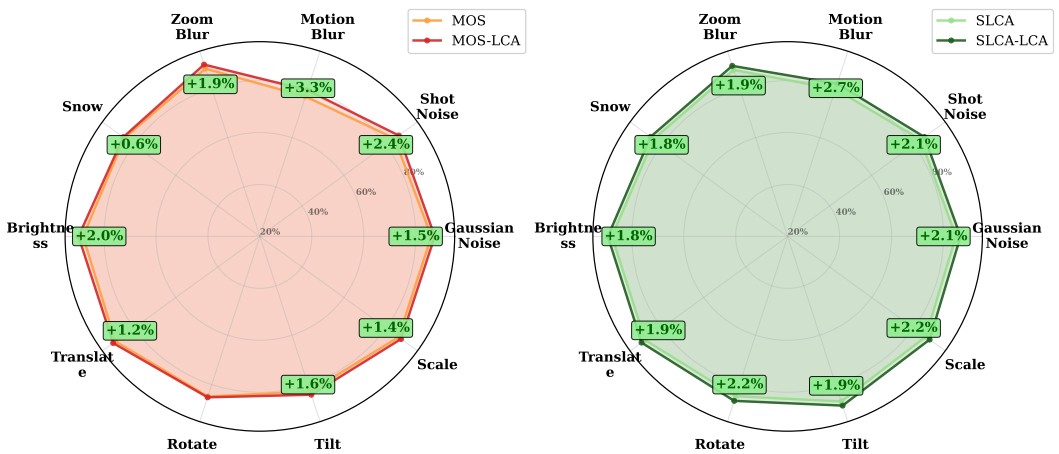

Figure 12: Accuracy of the original methods and their LCA variants on CIFAR100-P perturbations.

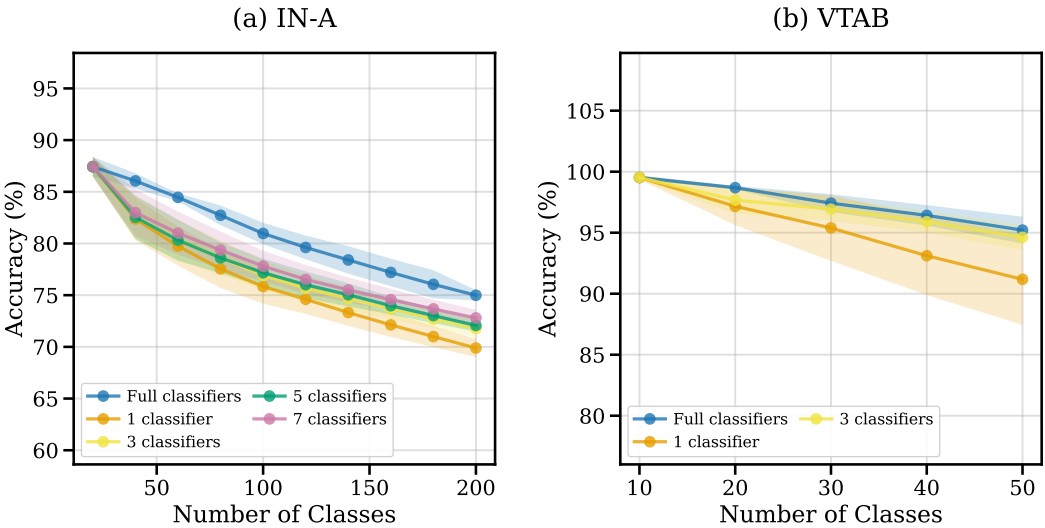

Figure 13: Accuracy of selectively finetuning a subset of classifiers.

