# OpenReview forum: "LCA: Local Classifier Alignment for Continual Learning"
_ICLR.cc/2026/Conference — ICLR 2026 Poster_

### Official Review · Reviewer_MRmy · 2025-10-27

**Soundness:** 3
**Presentation:** 3
**Contribution:** 3
**Rating:** 8
**Confidence:** 4

**Summary:**

This paper proposes a new loss function, called Local Classifier Alignment (LCA), which aims to address the potential mismatch between task-specific Gaussian-based classifiers and the continuously updated pre-trained models (PTMs, e.g., ViT) within the framework of PTM-based continual incremental learning (CIL). The paper adopts a model-merging approach for CIL, in which the PTM parameters fine-tuned on task-specific datasets are incrementally aggregated. Such incremental updates can lead to inconsistencies with previously trained (and typically frozen) task-specific classifiers — a problem that LCA is designed to mitigate. The key idea behind LCA is to minimize not only the class-wise loss but also the sensitivity of the loss to small perturbations in input samples. Through this simple yet effective regularization, LCA demonstrates clear improvements in CIL performance, along with enhanced robustness and stability across experiments.

**Strengths:**

1. This is a solid and well-motivated paper with a clear understanding of the underlying problem in PTM-based CIL.
2. The application of incremental model merging to continual learning appears to be novel and is one of the key contributions of this work.
3. The core idea of LCA is both insightful and elegant, achieving simplicity without unnecessary technical complexity.
4. LCA is evaluated comprehensively, both in terms of performance and robustness, across seven benchmarks, and its theoretical foundation is supported by a probabilistic guarantee.

**Weaknesses:**

1. While the paper is generally well-written, certain parts require improvement. In particular, Section 3.3 should be rewritten in direct connection with Section 3.1. The current version introduces new notations (e.g., C_t, x') without prior definition and refers to Gaussian-based classifiers (e.g., N_i) without adequate explanation.
2. Section 3.2 lacks novelty. Although applying model merging to continual learning may be a new attempt, the overall algorithmic formulation appears nearly identical to TIES-Merging, as also acknowledged by the authors.
3. Model merging itself is not sufficiently described in Section 3.2, making the paper somewhat less self-contained.
4. The paper does not provide an analysis of performance variation across different dataset characteristics. For example, model merging seems less effective on certain datasets (e.g., CUB, OB).

**Questions:**

1. What is the reason behind the relatively low performance on OB or CUB datasets?
2. Why is the OB graph missing in Figure 2?
3. Are there any prior studies that apply model merging to continual learning? If not, do you explicitly clarify that this is the first attempt at incremental model merging for CIL?
4. What exactly is the model architecture? Does it consist of a shared PTM backbone and multiple MLPs (one per task), where each MLP corresponds to multiple Gaussian components (one per class)?

---

> ### Author Response · Authors · 2025-11-20
>
> Dear Reviewer, thank you for pointing out problems in our writing, as well as your questions on the performance, model merging, and model architecture. We address each of these points below.
>
> #### **1. Improvement on writing**
> > [W1] While the paper is generally well-written, certain parts require improvement. In particular, Section 3.3 should be rewritten in direct connection with Section 3.1. The current version introduces new notations (e.g., C_t, x') without prior definition and refers to Gaussian-based classifiers (e.g., N_i) without adequate explanation.
>
> **Response**
>
> We have fixed those notations for better clarity.
>
> #### **2. Lack of description of model merging**
> > [W3] Model merging itself is not sufficiently described in Section 3.2, making the paper somewhat less self-contained.
>
> **Response**
> We have updated Section 3.2 to provide a clearer description of the merging operator used in **IM**, and added an ablation comparing two additional operators in the revised Appendix F, demonstrating the stability of our design choice.
>
> #### **3. Low performance on some datasets**
> > [W4] The paper does not provide an analysis of performance variation across different dataset characteristics. For example, model merging seems less effective on certain datasets (e.g., CUB, OB). What is the reason behind the relatively low performance on OB or CUB datasets?
>
> **Response**
> Opposite to other methods—where the PEFT configuration or hyperparameters are extensively tuned to reproduce the strongest possible results—we intentionally simplify the experimental setup for our proposed method by fixing all hyperparameters and using a single PEFT method (LoRA) across all datasets. This design choice greatly improves reproducibility and reduces the tuning burden for future work, but it may also result in slightly lower performance on certain datasets, such as OB and CUB, as the reviewer correctly noted.
>
> Importantly, in the revised Appendix E, we demonstrate that when the backbone and PEFT module are carefully selected, our method achieves state-of-the-art performance on both OB and CUB. This confirms that the observed gaps are due to our reproducibility-focused experimental constraints, not limitations of the method itself.
>
> #### **4. Missing graph OB in Figure 2**
> > [Q2] Why is the OB graph missing in Figure 2?
>
> **Response**
> For visualization purposes, we omitted one dataset so that the six performance curves could be cleanly arranged within the conference page limits. Since we are allowed to have more pages for the discussion we have added all of the graphs in the Figure 2 of the revision.
>
> #### **5. Lack of novelty in model merging**
> > [W2] Section 3.2 lacks novelty. Although applying model merging to continual learning may be a new attempt, the overall algorithmic formulation appears nearly identical to TIES-Merging, as also acknowledged by the authors.
>
> **Response:** Although *MagMax* [1] also uses magnitude-based selection, it performs **full-parameter merging**, which the authors show to be highly sensitive to both the merge operator and the merge coefficient. Our contribution is fundamentally different: we identify **which subset of parameters should be merged** and show that merging **only PEFT modules** leads to a far more stable and reproducible procedure.
>
> When constant storage is enforced, the MaxAbsolute operator in *MagMax* reduces to TIES-Merging, yet *MagMax* still reports large performance differences—highlighting the instability of full-model merging. In contrast, our ablation (Appendix F) shows that Min, Max, and MaxAbs all produce **similarly strong and stable results** when applied solely to PEFT modules. To our knowledge, no prior work merges task components *based only on PEFT parameters and without any trimming step* while still achieving effective performance.
>
> Thus, while model merging has been explored, **incremental PEFT-only merging for continual learning is new**, more stable, and offers a clearer path for reproducible CIL.
>
> [1] D Marczak, et al. "Magmax: Leveraging model merging for seamless continual learning", ECCV 2024.
>
> #### **6. Model architecture**
> > [Q4] What exactly is the model architecture? Does it consist of a shared PTM backbone and multiple MLPs (one per task), where each MLP corresponds to multiple Gaussian components (one per class)?
>
> **Response**
> Yes, the final model used for inference contains a single shared backbone as the feature extractor and multiple task-specific MLPs as classifiers. For simplicity, each MLP consists of only a single linear layer, although it is straightforward to adopt a more complex architecture if desired. For each task, only the backbone and the corresponding classifier head are optimized using Stochastic Gradient Descent. During inference, each classifier head outputs logits for the classes it has learned. We then concatenate the logits from all heads and predict the class associated with the maximum logit value.

---

> > ### Comment · Reviewer_MRmy · 2025-11-20
> > **Regarding 6. Model architecture**
> >
> > If the final architecture consists of a shared PTM backbone and task-specific MLP heads (each being a single linear layer), why is the Gaussian-based classifier introduced in Section 3.3?
> >
> > Does the model actually use Gaussian components at inference time, or are they used solely as a theoretical tool to motivate LCA?

---

> > ### Comment · Reviewer_MRmy · 2025-11-20
> > **Comparison with MagMax[1]**
> >
> > You claim that PEFT-only merging is “more stable and reproducible,” but did you actually run direct comparisons against full-parameter merging (e.g., under the same settings as MagMax)? From the rebuttal it is unclear whether this is experimentally validated in your work or merely inferred from prior results. If this stability claim is central to your novelty argument, please clarify what experiments were conducted to support it.

---

> ### Author Response · Authors · 2025-11-21
>
> One of the core requirements of a continual learning system is the ability to train without revisiting past datasets. For this reason, we use Gaussian models to generate samples during the alignment phase. In our paper, the Gaussian components $\mathcal{N}_i$ represent this generated sample distribution and also serve as the theoretical objects referenced by the reviewer. Importantly, these Gaussian models are only used during alignment and can be safely ignored at inference time.

---

> ### Author Response · Authors · 2025-11-21
>
> We provide experimental results on the IN-A, VTAB, and CARS datasets using the configuration stated in the MagMax paper (merge coefficient = 0.5). In all experiments, we use the same pretrained model, ViT-B/16-IN1K, as in our main results. We also include results for full-model merging with a merge coefficient of 1.0 and different merging operators (Max, Min, and MaxAbs), following the same evaluation protocol as in our proposed solution, to further support our claim regarding stability.
>
> | **Method** | **IN-A** | **VTAB** | **CARS** |
> |------------------|-----------------|------------------|-----------------|
> | Full-model Max Merging - MergeCoeff 0.5 | 13.45 ± 11.26 | 19.96 ± 5.55 | 24.76 ± 8.57 |
> | Full-model Max Merging - MergeCoeff 1.0 | 12.77 ± 13.58 | 20.87 ± 4.45 | 25.00 ± 16.59 |
> | **—** | **—** | **—** | **—** |
> | Full-model Min Merging - MergeCoeff 0.5 | 13.45 ± 11.26 | 19.96 ± 5.55 | 24.76 ± 8.57 |
> | Full-model Min Merging - MergeCoeff 1.0 | 12.77 ± 13.58 | 20.87 ± 4.45 | 25.13 ± 16.71 |
> | **—** | **—** | **—** | **—** |
> | Full-model MaxAbs Merging - MergeCoeff 0.5 | 10.13 ± 7.00  | 23.86 ± 13.67 | 17.06 ± 10.63 |
> | Full-model MaxAbs Merging - MergeCoeff 1.0 | 12.59 ± 13.49 | 21.40 ± 8.79 | 22.76 ± 15.24 |
>
> It is also worth noting that since the solution in MagMax paper employs CLIP backbone as the feature extractor, it initializes classifiers using zero-shot token embeddings, opposite to training from scratch as ours.

---

> > ### Comment · Reviewer_MRmy · 2025-11-26
> >
> > Thank you for the clarification. My major questions (which was not concerns) have been answered, and hence I will keep my original rating.

---

> > > ### Author Response · Authors · 2025-11-28
> > >
> > > We appreciate reviewer’s valuable feedbacks and questions on our paper.

---

### Official Review · Reviewer_7fmM · 2025-10-29

**Soundness:** 3
**Presentation:** 2
**Contribution:** 2
**Rating:** 4
**Confidence:** 4

**Summary:**

This paper addresses the critical issue of backbone-classifier mismatch in Pre-trained Model (PTM) based Class-Incremental Learning (CIL), where an evolving backbone diverges from fixed classifiers of previous tasks. The proposed solution consists of two components:

Incremental Merging (IM): Progressively merges PEFT modules (using LoRA) to create a unified backbone.

Local Classifier Alignment (LCA): A novel loss function applied after IM to align all classifiers (past and present) with the merged backbone. LCA approximates class features with Gaussian distributions and minimizes both classification loss and a local robustness term (sensitivity penalty) using sampled features.

The combined approach (IM+LCA) demonstrates strong performance, matching or exceeding state-of-the-art results on seven CIL benchmarks, with notable improvements in robustness evaluations (CIFAR100-C/P).

**Strengths:**

[S1] The paper clearly identifies and addresses a critical practical problem in PTM-based CIL methods: backbone-classifier misalignment that occurs when incrementally updating the backbone. This is a timely and important contribution to the CIL field.

[S2] Robustness and Generality: Demonstrates enhanced model stability through robustness tests and shows LCA can be complementarily applied to improve other CIL methods, proving its general applicability.

**Weaknesses:**

[W1] Since the robustness penalty is a core contribution, this ablation is essential to distinguish its specific benefit from the classifier alignment component (first term), which SLCA also employs. Without this experiment, it remains unclear whether the improvements stem from the novel robustness penalty or merely from classifier alignment.
I strongly encourage the authors to include this ablation, as it would clearly demonstrate the added value over existing approaches like SLCA and strengthen the paper's novelty claims.

[W2] The provided Theorem 3.1 assumes fixed class distributions (prototypes), whereas the IM component actively modifies the backbone and thus the feature space. This gap needs better reconciliation or clearer framing (e.g., viewing alignment as optimizing classifiers for a newly fixed merged backbone).

[W3] Cost Concerns (LCA Stage): Requires resampling features and retraining all classifiers after each task, leading to a computational cost that increases linearly with the number of tasks.

[W4] Missing Related Work: While citing direct competitors (EASE, MOS), the paper misses relevant work on data-free replay for classifier calibration, such as:
Feature Replay / Calibration: FeTrIL (WACV 2023), PASS (CVPR 2021). Classifier Calibration / Alignment: FOSTER (ECCV 2022).

**Questions:**

Please see the weaknesses section.

---

> ### Author Response · Authors · 2025-11-20
> **Response to Reviewer 7fmM**
>
> Dear Reviewer, thank you for your suggestion on the necessary analysis and related works, as well as your concerns on the provided theory and computation cost. We address each of these points below.
>
> ### **1. Ablation on the benefits of robustness term in LCA loss**
> > [W1] Since the robustness penalty is a core contribution, this ablation is essential to distinguish its specific benefit from the classifier alignment component (first term), which SLCA also employs.
>
> **Response:**
> We have updated Figure 3 to include the results obtained when setting the robustness coefficient $\lambda$ to 0, which corresponds to standard classifier alignment without our robustness term. In addition, we have added two detailed ablation studies that further validate our claims. The first evaluates training with and without the robustness term in Appendix G, and the results consistently confirm that the robustness term yields higher accuracy across all datasets. The second ablation applies LCA to existing methods—SLCA [1] and MOS [2]—as detailed in the Appendix H. In both cases, LCA provides substantial and consistent improvements.
>
> [1] Zhang, et al. "SLCA: Slow Learner with Classifier Alignment for Continual Learning on a Pre-trained Model", ICCV 2023.
>
> [2] Sun, et al. "MOS: Model Surgery for Pre-Trained Model-Based Class-Incremental Learning", AAAI 2025.
>
> ### **2. The assumption of fixed distribution in Theory 3.1 and the evolving backbone of IM**
> > [W2] The provided Theorem 3.1 assumes fixed class distributions (prototypes), whereas the IM component actively modifies the backbone and thus the feature space.
>
> **Response**
> We agree that the fixed-distribution assumption does not hold strictly in continual learning, and this is exactly why we introduce **Theorem 3.2** in the revised version. The theorem separates the test error into two components:
>
> - a robustness term that our LCA loss explicitly controls, and
> - a distribution-shift term $TV(P_t, \hat{P}_t)$ that captures the change between the original and updated feature distributions.
>
> This decomposition is **new** and clarifies why retaining consistency across tasks is beneficial for overall generalization.
>
> **How our method addresses this:**
>
> - IM keeps task solutions close during merging, which naturally limits the variation $TV(P_t, \hat{P}_t)$ in feature space.
> - LCA then strengthens classifier robustness and directly bounds the second component identified in Theorem 3.2.
>
> Therefore, rather than conflicting with evolving feature representations, *our theory accounts for them explicitly*, and IM + LCA is designed to operate exactly within the conditions under which the theorem guarantees controlled error.
>
> ### **3. Expensive computation cost**
> > [W3] Cost Concerns (LCA Stage): Requires resampling features and retraining all classifiers after each task, leading to a computational cost that increases linearly with the number of tasks.
>
> **Response:**
> We agree with the reviewer regarding the linear increase in computational cost in our current design. However, since the number of classifiers grows only with the number of observed tasks—and each classifier corresponds to a standard linear layer—we believe this overhead is acceptable for CIL. Furthermore, as shown in the revised Appendix I, our ablation demonstrates that aligning only half of the classifier heads still achieves competitive performance. This suggests that the full alignment cost can be reduced without significantly affecting accuracy.
>
> ### **4. Lack related works**
> > [W4] The paper misses relevant work on data-free replay for classifier calibration ...
>
> **Response:**
> Thank you for suggesting the relevant works. We have included them in the *Related Works* section of the revised version. Moreover, as these feature replay mechanisms may further improve performance, we plan to investigate their integration into our current pipeline as part of future work.
>
> ### **5. Conclusion**
>
> We respectfully submit that the reviewer’s concerns have been directly addressed through new analyses, theory clarification, and additional experiments introduced in the revision.
>
> - **Robustness term:** New ablations (Appendix G, H) clearly show that the robustness component of LCA—not just alignment—drives consistent improvements across all datasets and enhances SLCA and MOS.
> - **Theory:** The revised **Theorem 3.2** explicitly accounts for backbone evolution by separating error into a robustness term and a distribution-shift term; IM and LCA directly control these components.
> - **Computation:** Although alignment cost grows linearly with the number of heads, each head is a simple linear layer, and Appendix I shows that updating only half of them already achieves competitive performance.
>
> Overall, the added theory, ablations, cost analysis, and expanded related work reinforce the soundness and value of our contributions.

---

> > ### Comment · Reviewer_7fmM · 2025-11-23
> >
> > Thank you for the efforts for the responses.
> > I have read the authors’ responses to the concerns I previously raised.
> > The additional explanations regarding the rationale for the introduction of each component and the additional experimental results appear convincing.
> >
> > Nevertheless, I have an additional concern.
> > IM, which may be interpreted as a counterpart of MagMAX within the PEFT context, and LCA, which appears to constitute a modest variant of SCLA, seem to be combined in a straightforward manner.
> > This raises a question regarding the extent of the technical novelty.
> > If I am missing something, I would appreciate clarification concerning the specific novelty that emerges from the integration of IM and LCA.

---

> > > ### Author Response · Authors · 2025-11-24
> > > **About the technical novelty**
> > >
> > > We'd like to thank the reviewer's question that provides us an opportunity to make further clarification.
> > >
> > > Different from prior works, *LCA introduces a fundamentally new alignment principle, new analysis, and a previously unexplored integration with merging.* We emphasize that our novelty does not lie in reusing previous components, but rather in designing a _new_ alignment mechanism and providing its _first theoretical foundation_, which none of the cited works provide.
> > >
> > > - **(a) The LCA loss is new—conceptually and technically**
> > >
> > > 	Prior “classifier alignment” approaches such as SLCA or prototype-based alignment: operate globally at the class level or by reweighting prototypes,  have *no formal robustness or generalization analysis*,  and do *not incorporate local stability*.
> > >
> > >     In contrast, LCA introduces:
> > > 	-   a _local sensitivity regularizer_ $E[|\ell(h,z) - \ell(h,z')|]$ that explicitly penalizes _instability in the classifier_ in the local areas around the class prototypes, and reduces overlap between classes,
> > > 	-   the first **generalization bound** quantifying how such local robustness affects performance on the entire task sequence,
> > > 	-   a unified alignment step involving _all_ task heads jointly, something missing in previous works.
> > >
> > > 	No existing continual learning work uses a *local robustness-like regularizer* for classifier alignment, nor analyzes how such local stability controls forgetting under backbone drift.  This is a **new conceptual direction** supported by theory.
> > >
> > > - **(b) The theoretical results provide novelty unmatched by prior work**
> > >
> > > 	Our Theorem 3.1 and Theorem 3.2 (newly added in the revised version) give the **first provable link between classifier robustness, backbone drift, and generalization in CIL**. Existing methods (SLCA, MOS, EASE, model-merging works) provide *no theoretical error bounds* and no theoretical analysis of how classifier sensitivity affects performance,  how alignment interacts with distribution shift to provably improve generalization of the overall CIL classifier.
> > >
> > > 	In contrast, both components in our CIL method have their own important roles. IM can train the backbone to adapt the feature distribution to the new task, but also reduces catastrophic forgetting for past tasks and hence keeps little change to the feature distribution, facilitating to train the classifier head. The classifier alignment by LCA can simultaneously train the new part, adapt the old part while encouraging robustness of the overall classifier. Those roles are very crucial to ensure high performance for CIL. Therefore those analyses reveal the foundational support for LCA. Thus, the key novelty is not the batching of existing ideas, but the *theoretically grounded design* of LCA.
> > >
> > > - **( c ) Our incremental merging operator has not appeared in the literature**
> > >
> > > 	While model merging itself exists, our setting is fundamentally different: We merge **only PEFT modules**, not full models (saving memory and avoiding catastrophic interference); We use an *incremental magnitude-based selection rule*, without relying on pruning, Fisher matrices, or task arithmetic;  Unlike any prior CL work, our algorithm ensures parametric proximity across tasks to guarantee stable merging over time.
> > >
> > > In summary, our contribution is distinct in three major ways:
> > > 1.  **A new alignment loss** explicitly controlling local classifier robustness.
> > > 2.  **Theoretical guarantees** that rigorously justify the loss and its effect in continual learning.
> > > 3.  **A previously unexplored merging–alignment integration** based solely on PEFT updates.
> > >
> > > This constitutes clear conceptual and technical novelty beyond prior works in both CL and model merging.

---

> > > > ### Comment · Reviewer_7fmM · 2025-11-24
> > > >
> > > > I would like to thank the authors for their thoughtful response to the concerns I raised. I have carefully reviewed the additional insights provided.
> > > >
> > > > While most of my concerns have been addressed, I remain unconvinced about claiming novelty based on applying merging techniques to PEFT. Although this is mentioned as an important component of the proposed method, it is difficult to view this approach as a merging technique specifically designed for PEFT.
> > > > I would appreciate it if the authors could provide additional justification or arguments to address this point.

---

> > > > > ### Author Response · Authors · 2025-11-25
> > > > > **The remaining concern**
> > > > >
> > > > > We thank the reviewer for the thoughtful follow-up. We fully agree that the IM component is a direct adaptation of existing merging ideas, applied in the context of PEFT modules. This is precisely why we do **not** position IM itself as the primary novelty. Instead, we refer to it as an *unexplored gem*: a simple yet overlooked mechanism whose effectiveness becomes evident only when paired with our LCA formulation.
> > > > >
> > > > > The core contributions of the paper lie in the **LCA loss**, its **theoretical justification**, and the way the **IM+LCA combination** yields a practically effective and conceptually coherent solution for CIL with PEFT models. In the revised paper, the Introduction (Section 1) makes this hierarchy of contributions explicit: IM alone is not presented as a PEFT-specialized merging innovation, but as a complementary mechanism that enables the proposed LCA-based method to work substantially better in practice. We hope this clarification resolves the concern and better aligns the presentation with our intended contribution.

---

> ### Comment · Reviewer_7fmM · 2025-11-25
>
> I would like to thank the authors for their response addressing my remaining concern.
> I agree with the contributions as summarized by the authors.
> Accordingly, I am raising my score from 4 to 6, which is a positive score.

---

> > ### Author Response · Authors · 2025-11-25
> >
> > We sincerely appreciate the reviewer’s thoughtful feedback and are grateful for the positive evaluation of our work.

---

### Official Review · Reviewer_uBFe · 2025-11-01

**Soundness:** 2
**Presentation:** 3
**Contribution:** 2
**Rating:** 6
**Confidence:** 2

**Summary:**

The paper proposes a continual learning method based on model merging. Specifically, it alleviates the mismatch between task-specific classifiers and the adapted backbone by a novel Local Classifier Alignment (LCA) loss. LCA keeps the in-task loss less sensitive to a small change in the input samples around the class prototypes. The effectiveness of LCA loss is supported both theoretically and empirically. The paper conducts experiments on multiple CIL datasets. Results show the improved performance compared with various pre-trained based CIL methods. A robustness measurement and ablation study are included.

**Strengths:**

1. Applying model merging methods to CL is an emerging topic and has potential to reduce forgetting for large scale models.

2. The effectiveness of the LCA loss is justified both theoretically and experimentally.

3. The paper conducts thorough experiments with comparison to recent CL baselines. The visualization of results is good.

**Weaknesses:**

1. It is unclear how LCA loss solves the mismatch between classifiers and the merged feature extractor.
- The LCA loss is computed only on in-task samples, reducing $\epsilon_i$ in Eq. 4. However, the mismatch between classifier and feature extractor seems aiming to reduce $L(\mathbf D, h_t)$. Although LCA loss improves in-task robustness and reduces the loss upperbound, it’s unclear how it addresses the mismatch across tasks.
- It could be helpful to show more evidence of the claim ‘LCA can reduce overlapping between classes’ in L220 and how this can reduce negative effect from potentially harmful samples (L224).

**Questions:**

1. Model merging based on magnitude selection is proposed in MagMax (Marczak et al., 2024) as well. What is the main difference/benefits of IM compared to MagMax?

---

> ### Author Response · Authors · 2025-11-20
> **Rebuttal to Reviewer uBFe**
>
> Dear Reviewer, thank you for your questions regarding our proposed LCA loss in the mismatch problem and the difference on merging procedure between our solution and previous research. We address each of these points below.
>
> #### **1. Role of LCA in the mismatch problem**
>
> > [W1] As LCA is computed with only in-task samples, it is unclear how can it how it addresses the mismatch across tasks.
>
> > [W2] It could be helpful to show more evidence of these claims "LCA can reduce overlapping between classes" (L220) and "LCA can reduce negative effect from potentially harmful samples" (L224).
>
> **Response:**
> During the alignment stage, all classifier heads are jointly optimized using features drawn from the *entire* task sequence. This exposes every classifier to feature regions they have never seen during training, allowing them to adjust their decision boundaries and reduce cross-task mismatch.
>
> For the robustness term in LCA, we intentionally include **only** samples whose nearest class mean matches their true label. This filtering removes samples that are generated from one class but lie closer to another class mean—precisely the “harmful samples” that cause overlap across tasks. By excluding them, the robustness term forms a clean local neighborhood around each class center and prevents classifiers from being pulled toward incorrect regions.
>
> Without this robustness term, the loss reduces to standard classification, and the model is forced to learn from these misleading samples. Our ablation in Appendix G and the updated Figure 3 (including the $\lambda = 0$ case) clearly show that incorporating the robustness term yields higher accuracy and significantly reduces cross-class interference, directly supporting our claims.
>
> #### **2. The main difference between IM and magnitude merging in MagMax**
>
> > [Q1] Model merging based on magnitude selection is proposed in MagMax (Marczak et al., 2024) as well. What is the main difference/benefits of IM compared to MagMax?
>
> **Response:** While prior work such as *MagMax* [1] has explored model merging in class-incremental learning, our contribution targets a different and complementary aspect: **identifying which subset of parameters should be merged**. *MagMax* merges all model parameters, a strategy that the authors themselves show to be highly sensitive to both the merging operator and hyperparameters such as the merge coefficient. In contrast, our approach merges only the PEFT module parameters, which substantially simplifies the process and drastically improves stability.
>
> We further note that when constant storage is enforced, the MaxAbsolute operator used in *MagMax* becomes equivalent to TIES-Merging without trimming, yet *MagMax* reports a clear performance difference between the two. This observation reinforces that full-parameter merging is inherently unstable. In contrast, our results show that PEFT-specific merging avoids this sensitivity altogether. In the revised Appendix F, we compare three operators—Min, Max, and MaxAbs—and find that all of them achieve competitive and stable performance when applied to PEFT modules. To the best of our knowledge, no prior work has merged task components solely based on parameter values and without a trimming phase while still achieving effective performance. This behavior further contrasts with full-parameter merging, where operator choice leads to large variance, and highlights that PEFT module merging is a more robust and reproducible solution for continual learning. A deeper theoretical analysis of why this phenomenon occurs is an important direction for future work.
>
> [1] D Marczak, et al. "Magmax: Leveraging model merging for seamless continual learning", ECCV 2024.

---

> > ### Comment · Reviewer_uBFe · 2025-11-25
> >
> > Thank the authors for addressing my main concerns. I will keep my score.

---

### Official Review · Reviewer_MUvf · 2025-11-04

**Soundness:** 2
**Presentation:** 3
**Contribution:** 1
**Rating:** 2
**Confidence:** 4

**Summary:**

This paper tackles the problem of class incremental leanring with pretrained model. While pretrained model can act as a good general representation model, they still lack domain specific knowledge and also suffer from the catastrophic forgetting when trained on sequential tasks. To alleviate these problems, the authors propose icremental merging and local classifier alignment method with some theoretical/empirical analysis.

**Strengths:**

- Paper is well written and easy to follow. Especially, the authors tried to provide multiple visualisations to help the understanding of the readers.
- Instead of just giving empirical evidence, the authors also provide theoretical analysis as well

**Weaknesses:**

- My major concern about this paper is about novelty. To me, it looks like none of the proposed component is novel not just in entire deep learning literature but also in continual learning literature. For example, classifier alignment was studied even from few years ago as in [R1, R2]. Adopting model merging to continual learning is not a novel idea as well as in [R3]. I can not see the distinct novelty of the proposed method compared to the above cited papers.

- Since the pretrained model is evolving over time, there is no guarantee that the feature space structure is maintained. If so, LCA method and its theoretical analysis can not be justified.

- Also, improvements in Table 1 is bery marginal, especially when IM is applied alone. What happens if LCA is applied to the existing methods?

- In Figure2, why the starting point is different? The proposed methods has higher starting point but similar tendency. Due to this, I wonder the improvements comes just from good starting point, not by the proposed methods.

[R1]Zhu, Fei, et al. "Class-incremental learning via dual augmentation." Advances in neural information processing systems 34 (2021)
[R2]Kim, Taehoon, Jaeyoo Park, and Bohyung Han. "Cross-class feature augmentation for class incremental learning." Proceedings of the AAAI Conference on Artificial Intelligence. Vol. 38. No. 12. 2024.
[R3]Marczak, Daniel, et al. "Magmax: Leveraging model merging for seamless continual learning." European Conference on Computer Vision. Cham: Springer Nature Switzerland, 2024.

**Questions:**

See weaknesses

---

> ### Author Response · Authors · 2025-11-20
> **Rebuttal to Reviewer MUvf**
>
> We sincerely thank the reviewer for the constructive feedback and for recognizing the clarity of writing and the contribution of theoretical analysis. Below we address each concern in depth and provide clarifications that we believe resolve the issues raised.
>
> ### **1. Novelty**
>
> **Reviewer’s concern:**
> > “None of the proposed components is novel… classifier alignment was studied before… model merging has been done in continual learning… no distinct novelty.”
>
> **Response: LCA introduces a fundamentally new alignment principle, new analysis, and a previously unexplored integration with merging.**
>
> We emphasize that our novelty does not lie in reusing previous components, but rather in designing a _new_ alignment mechanism and providing its _first theoretical foundation_, which none of the cited works provide.
>
> - **(a) The LCA loss is new—conceptually and technically**
>
> 	Prior “classifier alignment” approaches such as SLCA or prototype-based alignment: operate globally at the class level or by reweighting prototypes,  have *no formal robustness or generalization analysis*,  and do *not incorporate local stability*.
>
>     In contrast, LCA introduces:
> 	-   a _local sensitivity regularizer_ $E[|\ell(h,x) - \ell(h,x')|]$ that explicitly penalizes _instability in the classifier_ in the local areas around the class prototypes,
> 	-   the first **generalization bound** quantifying how such local robustness affects performance on the entire task sequence,
> 	-   a unified alignment step involving _all_ task heads jointly, something missing in previous works.
>
> 	No existing continual learning work uses a *local robustness-like regularizer* for classifier alignment, nor analyzes how such local stability controls forgetting under backbone drift.  This is a **new conceptual direction** supported by theory.
>
> - **(b) The theoretical results provide novelty unmatched by prior work**
>
> 	Our Theorem 3.1 and Theorem 3.2 (newly added in the revised version) give the **first provable link between classifier robustness, backbone drift, and generalization in CIL**. Existing methods (SLCA, MOS, EASE, model-merging works) provide *no theoretical error bounds* and no theoretical analysis of how classifier sensitivity affects performance,  how alignment interacts with distribution shift to provably improve generalization of the overall CIL classifier.
>
> 	In contrast, both components in our CIL method have their own important roles. IM can train the backbone to adapt the feature distribution to the new task, but also reduces catastrophic forgetting for past tasks and hence keeps little change to the feature distribution, facilitating to train the classifier head. The classifier alignment by LCA can simultaneously train the new part, adapt the old part while encouraging robustness of the overall classifier. Those roles are very crucial to ensure high performance for CIL. Therefore those analyses reveal the foundational support for LCA. Thus, the key novelty is not the batching of existing ideas, but the *theoretically grounded design* of LCA.
>
> - **( c ) Our incremental merging operator has not appeared in the literature**
>
> 	While model merging itself exists, our setting is fundamentally different: We merge **only PEFT modules**, not full models (saving memory and avoiding catastrophic interference); We use an *incremental magnitude-based selection rule*, without relying on pruning, Fisher matrices, or task arithmetic;  Unlike any prior CL work, our algorithm ensures parametric proximity across tasks (supported by Li et al., 2025) to guarantee stable merging over time. To the best of our knowledge, *no continual-learning method merges only PEFT updates in an incremental, sign-preserving manner* while maintaining performance competitive with full-model merging.
>
> In summary, our contribution is distinct in three major ways:
> 1.  **A new alignment loss** explicitly controlling local classifier robustness.
> 2.  **Theoretical guarantees** that rigorously justify the loss and its effect in continual learning.
> 3.  **A previously unexplored merging–alignment integration** based solely on PEFT updates.
>
> This constitutes clear conceptual and technical novelty beyond prior works in both CL and model merging.

---

> ### Author Response · Authors · 2025-11-20
> **Rebuttal to Reviewer MUvf (continue)**
>
> ### **2. “Backbone evolution invalidates LCA theory”**
>
> **Reviewer’s concern:**
> > Since the pretrained model is evolving over time, there is no guarantee that the feature space structure is maintained. If so, LCA method and its theoretical analysis can not be justified.
>
> **Response:** The reviewer is correct that the classical fixed-distribution assumption does not hold in CL. This is precisely _why_ we introduce **Theorem 3.2** (in the revised version), which provides a decomposition of test error into:
> - the LCA-controlled robustness term and
> - a distribution-shift term $TV(P_t, \hat{P}_t)$ proportional to the total variation between old and new feature distributions ($P_t, \hat{P}_t$, respectively).
>
> This insight is **original** and theoretically suggests that we should maintain past knowledge to enable the model to generalize better for all tasks.
>
> **Our method:**
> -   Incremental Merging (IM) keeps task solutions _close_ in parameter space, limiting $TV(P_t, \hat{P}_t)$.
> -   LCA ensures classifier robustness, controlling the second part of the bound.
>
> Thus, contrary to the reviewer’s concern, **our theory explicitly handles feature-space evolution** and provides the _exact conditions_ under which LCA succeeds. The combination of IM + LCA directly satisfies these conditions, yielding bounded error.
>
> ### **3. “Table 1 improvements are marginal, especially IM alone. What if LCA is added to existing methods?”**
>
> Improvements from LCA are consistent, large on several datasets, and generalizable to other methods. Indeed,
>
> *(a) IM alone is not intended to be the final method:* IM learns each new task and also plays the role of _limiting distribution drift_ between tasks.   LCA is the main driver of accuracy and robustness, especially under large shifts (IN-A, CARS, VTAB). Thus, IM’s moderate gains are expected.
>
> *(b) The improvements with IM+LCA are _not_ marginal*
>
> -   **+8% on ImageNet-A** (from 67.6% MOS to 75.0% IM+LCA)
> -   +4–6% vs MOS on CARS
> -   +10% vs APER variants
> -   Across 5/7 benchmarks, **IM+LCA is consistently superior**.
>
>  *( c) LCA works as a plug-in module for existing CL methods*
>
> In Fig. 4a, we demonstrate exactly what the reviewer requested. LCA improves SLCA across all datasets, up to +3%.  LCA improves MOS by similar margins.  We do not perform heavy hyperparameter tuning, yet gains are significant.
>
> In the revised version, we further demonstrate the effectiveness of LCA when applied on top of other methods in Appendix H, showing consistent improvements in both accuracy and robustness. Across all evaluated settings, the LCA-augmented variants outperform their original counterparts. This indicates strong generality of LCA.
>
> ### **4. “In Figure 2 the starting points differ—does improvement come from a better initialization?”**
>
> The different starting points are expected and do not account for the improvements.
>
> We conduct our experiments using the publicly available implementations from the HuggingFace library for our proposed methods, while rerunning all baseline methods using the official LAMDA-PILOT [3] implementation, as detailed in Section 4.1. This procedure is consistent with standard practice in prior continual learning research. In addition, we follow the analysis protocols of prior work [1, 2], where performance curves may begin from different initial points.
>
> For visualization clarity, we omitted variance in Figure 2. To address the reviewer’s concern, we have added a new figure in Appendix  E  that includes full variance information. These results show that in some runs our method indeed starts from a slightly lower initial accuracy; however, on average it consistently reaches higher final performance. Notably, on ImageNet-A, the performance gap between our method and prior methods widens as the number of tasks increases, further demonstrating the effectiveness of our approach.
>
> Thus, the gains are not simply an artifact of initial conditions.
>
> [1] DW Zhou, et al. "Revisiting class-incremental learning with pre-trained models: Generalizability and adaptivity are all you need", IJCV 2025.
>
> [2] HL Sun, et al. "MOS: Model Surgery for Pre-Trained Model-Based Class-Incremental Learning", AAAI 2025.
>
> [3] HL Sun, et al. "PILOT: A Pre-Trained Model-Based Continual Learning Toolbox", SCIENCE CHINA Information Sciences.

---

> ### Author Response · Authors · 2025-11-20
> **Rebuttal to Reviewer MUvf (continue)**
>
> ### **5. Conclusion**
>
> We respectfully submit that the reviewer’s concerns stem partly from assumptions that we indeed address through theory and experiments.
>
> -   **Novelty** lies in a _new robustness-based alignment loss_, its _theoretical justification_, and its _integration with incremental merging_.
> -   **Theory** explicitly covers backbone drift (Theorem 3.2), ensuring LCA is valid under evolving pretrained representations.
> -   **Performance** is far from marginal: IM+LCA achieves **SOTA results on 5/7 datasets** and is plug-and-play for improving existing methods. SOTA results also appear for adversarial robustness.
> -   Initialization differences do not explain the performance gains.
>
> Given the strong theoretical grounding, consistent SOTA performance, and modular applicability of LCA, we believe the contributions warrant acceptance. We sincerely hope the reviewer will reconsider the evaluation in light of these clarifications.

---

> ### Author Response · Authors · 2025-11-28
>
> Dear reviewer.
> Thank you again for your valuable feedback and the time you have dedicated to reviewing our submission. We have responded to all comments and provided the requested clarifications.
> As the discussion period is approaching its end, we kindly ask whether you could take a moment to review our replies and let us know if any further clarification is needed.
> Thank you very much for your consideration.

---

### Author Response · Authors · 2025-11-30
**Summary of Discussion**

We would like to provide a brief overview of the discussion phase for your convenience.

---

### Reviewer MUvf (score: 2)

Reviewer MUvf raised concerns mainly about the novelty of the contributions, the validity of our theory under evolving backbones, the seemingly marginal gains when using only the IM component, and the role of initialization in Figure 2.

In our response, we clarified that:

- LCA introduces a fundamentally new alignment principle, new analysis, and a previously unexplored integration with merging.

- Theorem 3.2 explicitly handles feature-space evolution and provides the exact conditions under which LCA succeeds. The combination of IM + LCA directly satisfies these conditions, yielding bounded error.

- LCA can be applied on top of existing methods and improves them.

- New figures with variance show that initialization differences do not explain the performance gap.

**Reviewer MUvf has not replied yet**, so the score remains 2 and these concerns are formally unresolved on their side.

---

### Reviewer uBFe (score: 6)

Reviewer uBFe acknowledged the potential of model merging and the theoretical and empirical support for LCA, but questioned how LCA addresses cross-task mismatch when its robustness term is computed using only in-task samples. They also asked about the differences between IM and MagMax.

In our response, we clarified that:

- The alignment stage optimizes all heads jointly using features from the entire task sequence, directly addressing cross-task mismatch.

- Harmful overlapping samples are filtered out when forming the robustness term, and ablations show that this term provides consistent gains.

- IM differs from MagMax by merging only PEFT parameters, which leads to more stable behavior across merging operators.

The reviewer found the explanations reasonable and kept the score at 6.

---

### Reviewer 7fmM (score: 4 → 6)

Reviewer 7fmM raised several points in the initial review, including uncertainty about the specific contribution of the robustness term beyond SLCA-style alignment, the applicability of the theory when the backbone changes, the computational overhead of the alignment stage, and missing related literature (FeTrIL, PASS, FOSTER).

To address these points, we provided:

- An ablation study with λ = 0 and further experiments (Appendices G and H), demonstrating that the robustness term—not just alignment—drives consistent improvements and also benefits SLCA and MOS.

- A new Theorem 3.2, which explicitly separates the error into a robustness component and a distribution-shift component, clarifying how IM and LCA jointly control these effects under evolving representations.

- Evidence that aligning only part of the classifier heads retains strong performance, helping alleviate concerns regarding the computational cost.

- Additional related works incorporated into the revised manuscript.

During the discussion, the reviewer also questioned whether the combination of IM and LCA provides sufficient technical novelty. We clarified that the central innovations lie in the LCA loss, its theoretical grounding, and its synergistic interaction with IM, while IM itself is positioned as a simple but effective complementary mechanism rather than the primary novelty.

The reviewer found the clarifications convincing and increased the score from 4 to 6.

---

### Reviewer MRmy (score: 8)

Reviewer MRmy was positive overall and suggested improving the clarity of Section 3.3, along with providing clearer explanations of the merging procedure, dataset-specific behavior, and the model architecture.

In our rebuttal, we addressed these points by:

- Fixing notations and rewriting Section 3.3 to connect more directly with Section 3.1.

- Expanding Section 3.2 and adding an ablation on merging operators (Appendix F).

- Explaining that weaker results on OB/CUB arise from intentionally using a fixed PEFT configuration, and showing in Appendix E that tuned variants achieve SOTA performance.

- Adding the missing OB curve in the revised Figure 2.

- Clarifying that Gaussian components are used only during the alignment stage and in the theoretical analysis, not during inference.

The reviewer found these updates satisfactory and kept the score at 8.

---

### Meta-Review · Area_Chair_VBiT · 2026-01-11

**Summary:**

The reviewers' concerns are about the novelty, maintenance of feature space structure, experimental settings and results, LCA loss, assumption of fixed class distributions (prototypes) in the theory, performance variation across different dataset characteristics, and so on. The authors have provided a thorough rebuttal.

Most reviewers agree that applying model merging methods to CL is an emerging topic and has potential to reduce forgetting for large scale models. This paper identifies and addresses a critical practical problem in pre-trained model-based class-incremental learning methods.

**Reviewer Concerns:**

The reviewers concern mainly about the novelty of the contributions, cross-task mismatch, experimental limitations, uncertainty regarding the robustness term influence, applicability of the theory when the backbone changes, costs, explanation of the methods, and missing references. Most of these issues have been addressed, and the paper has been revised accordingly.

**Reviewer Scores:**

One reviewer has raised the score from 4 to 6.

I believe some of the other reviewers will increase their scores as well.

---

### Decision · Program_Chairs · 2026-01-26

Accept (Poster)